# MicroVerse: A Preliminary Exploration Toward a Micro-World Simulation

**Rongsheng Wang**[1,3]*  **Minghao Wu**[1]*  **Hongru Zhou**[2]  **Zhihan Yu**[1]
**Zhenyang Cai**[1]  **Junying Chen**[1]  **Benyou Wang**[1,3]†

[1]The Chinese University of Hong Kong, Shenzhen
[2]Peking Union Medical College Hospital
[3]Shenzhen Loop Area Institute

## Abstract

Recent advances in video generation have opened new avenues for macroscopic simulation of complex dynamic systems, but their application to microscopic phenomena remains largely unexplored. Microscale simulation holds great promise for biomedical applications such as drug discovery, organ-on-chip systems, and disease mechanism studies, while also showing potential in education and interactive visualization. In this work, we introduce **MicroWorldBench**, a multi-level rubric-based benchmark for microscale simulation tasks. MicroWorldBench enables systematic, rubric-based evaluation through 459 unique expert-annotated criteria spanning multiple microscale simulation task (e.g., organ-level processes, cellular dynamics, and subcellular molecular interactions) and evaluation dimensions (e.g., scientific fidelity, visual quality, instruction following). MicroWorldBench reveals that current SOTA video generation models fail in microscale simulation, showing violations of physical laws, temporal inconsistency, and misalignment with expert criteria. To address these limitations, we construct **MicroSim-10K**, a high-quality, expert-verified simulation dataset. Leveraging this dataset, we train **MicroVerse**, a video generation model tailored for microscale simulation. MicroVerse can accurately reproduce complex microscale mechanism. Our work first introduce the concept of **Micro-World Simulation** and present a **proof of concept**, paving the way for applications in biology, education, and scientific visualization. Our work demonstrates the potential of educational microscale simulations of biological mechanisms. Our data and code are publicly available at
**https://github.com/FreedomIntelligence/MicroVerse**

## 1 Introduction

World models LeCun (2022); Bruce et al. (2024); Lu et al. (2024) have been extensively studied for their ability to simulate environments and agent interactions. They offer a unified computational framework for perceiving surroundings, controlling actions, and predicting outcomes, thereby reducing reliance on real-world trials. This not only robotics engines  Luo & Du (2024); Lu et al. (2024) engines and reinforcement learning planners Hafner et al. (2020); Agarwal et al. (2025), but also enhances decision-making, supports safe exploration, and enables scalable learning.

Recently, video generative models have demonstrated strong potential to acquire commonsense knowledge directly from raw video data, ranging from physical laws in the real world to embodied behavioral patterns Brooks et al. (2024), laying the foundation for their use as real-world simulators. For example, prior work Luo & Du (2024) employs video-guided goal-conditioned exploration, grounding large-scale video generation model priors into continuous action spaces through self-supervision, enabling robots to master complex manipulation skills without explicit actions or rewards; and other works Lu et al. (2024) leverage video generation models for embodied decision-

---

*Equal Contribution.
†Corresponding author.

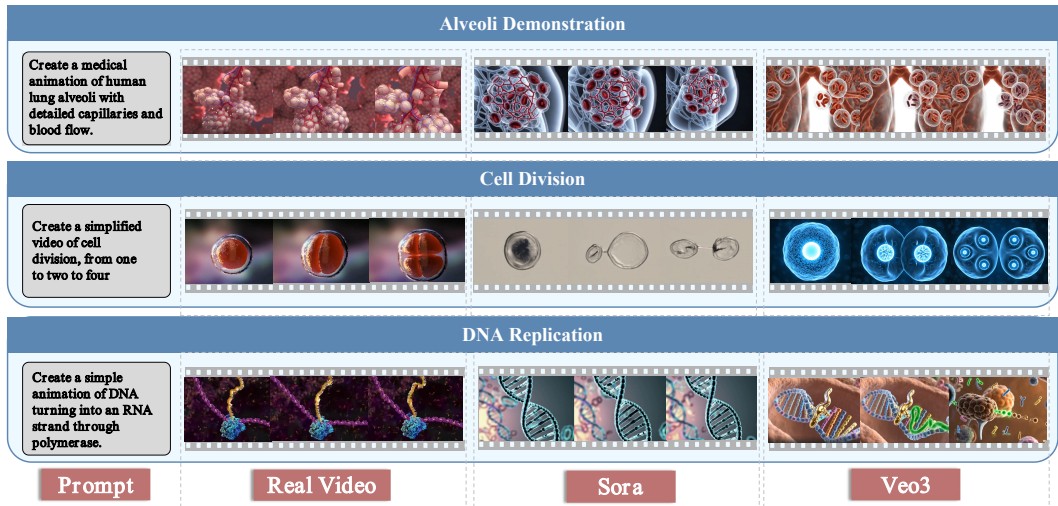

Figure 1: Failure cases of Sora and Veo3 on Microscale Simulation. Although Sora and Veo3 generate results that appear visually correct, their violations of physical laws are particularly evident.

making, allowing agents to imaginatively explore their environment with high generative quality and consistent exploration.

Despite tremendous progress in video generation for natural scenes and human-centered domains OpenAI (2024); Google DeepMind (2025); Kong et al. (2024); Wan et al. (2025); Yang et al. (2024), research efforts have remained predominantly focused on the macroscopic scale. This success has not translated effectively to the microscopic scale, where current state-of-the-art models fail to produce physically plausible or biologically meaningful dynamics, as shown in Figure 1. Microscopic simulation, which tracks the interactions of atoms, molecules, and cells to uncover underlying mechanisms, is crucial for applications in materials science, biomedical research Dario et al. (2000), education Romme (2002), and interactive visualization White (1992). The failure of existing models, primarily due to a lack of incorporated biomedical knowledge, highlights a critical gap despite the strong potential of microscale simulation for generating clinically realistic dynamics in fields like drug discovery and disease modeling. To address this, we aim to explore the potential of educational microscale simulations of biological mechanisms.

In this work, we introduce **MicroWorldBench**, a multi-level rubric-based benchmark for microscale simulation tasks comprising 459 real-world tasks that span organ-level, cellular, and subcellular processes. These tasks were jointly selected from a large candidate pool by LLMs and domain experts for their diversity and relevance, with each task paired with self-contained, objective evaluation criteria specifying the essentials for valid simulation. Our extensive experiments across a broad spectrum of video generation models reveal that while most maintain superficial visual coherence and adhere to prompts, they perform poorly in microscale settings, consistently failing to generate biologically plausible dynamics. These failures indicate that current models, trained predominantly on human-scale videos, lack grounding in microphysical principles and knowledge.

To mitigate the gap, we introduce **MicroVerse**, a video generation model tailored for microscale simulation. MicroVerse is built on Wan2.1 Wan et al. (2025) model and trained with **MicroSim-10K**, the first microscale dataset containing 9,601 expert-verified scenarios. Unlike human-scale datasets, MicroSim-10K emphasizes physical plausibility and biological fidelity across diverse microscale mechanisms. On MicroWorldBench, MicroVerse surpasses original model by more than +2.7 in scientific fidelity, highlighting the importance of domain-specific data.

Our contributions are summarized as follows: (i) We introduce the concept of **Micro-World Simulation** and present a **proof of concept**, which includes a clear objective, a dedicated benchmark, a training dataset, and a tailored model. (ii) We propose MicroWorldBench, the first rubric-based benchmark specifically designed for evaluating microscale simulation in video generation; (iii) we construct MicroSim-10K, a large-scale, expert-verified dataset of microscale simulation videos; (iv) We introduce MicroVerse , a fine-tuned video generation model built upon MicroSim-10K, achiev-

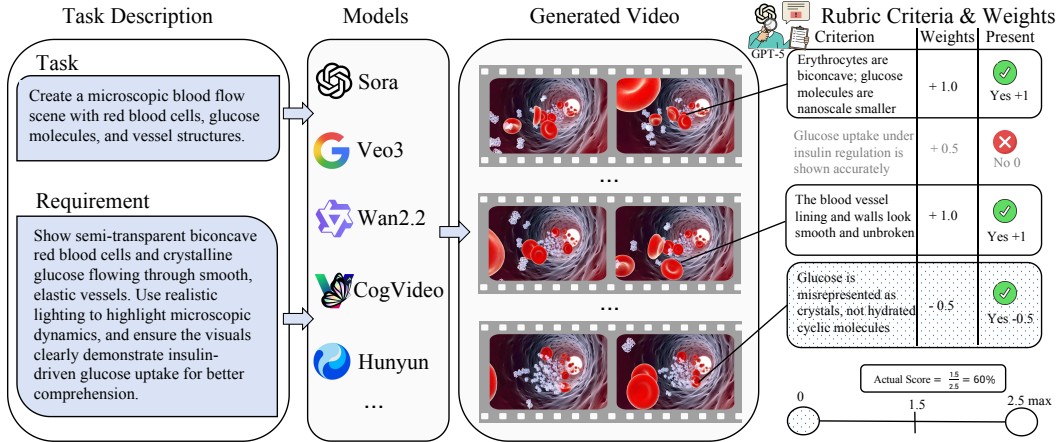

Figure 2: Illustration of MicroWorldBench Evaluation Process. A MicroWorldBench example consists of a generated microscopic video and a set of task-specific evaluation criteria written by experts. A MLLM-based scoring system rates responses according to each criterion.

ing competitive performance on MicroWorldBench by reducing violations of scientific constraints and improving temporal and spatial consistency.

# 2 MICROWORLDBENCH: A RUBRIC-BASED BENCHMARK FOR MICROSCALE SIMULATION

**Generic Evaluation Fails to Capture Microscale Simulation Dynamics** Existing evaluation methods for video models often rely on generic scoring rules or high-level principles Huang et al. (2024); Zheng et al. (2025); He et al. (2024), which are insufficient for microscale simulation. Such methods overlook the need for fine-grained microscopic simulations, resulting in misaligned outcomes and failing to capture deficiencies in physical plausibility and biological fidelity. In this work, the proposed Rubric evaluation addresses this gap by introducing task-specific criteria with differentiated weights. Rubrics highlight the most critical dimensions identified by experts and ensure that evaluations emphasize substantive shortcomings rather than being diluted by aggregate scoring.

In this section, we introduction the core structure of the rubric-based benchmark, covering task selection (Sec. 2.1), prompt design (Sec. 2.2), and rubric construction (Sec. 2.3), and describe the methodology for model evaluation (Sec. 2.4).

## 2.1 TASK CHOICE

Biological systems are inherently hierarchical, encompassing levels from society, body, organ, and tissue to cell, organelle, protein, and gene Qu et al. (2011). Given constraints of practicality impact and data availability, in this work we focus on three representative levels as a principled sampling of this hierarchy. Importantly, this choice does not discard existing scientific frameworks, but rather reflects a consensus-based selection of the most representative and tractable scales.

1. **Organ-level simulations** are essential because they connect microscale behaviors with macroscopic physiological functions. Dynamic processes such as cardiac contraction or vascular deformation are directly related to medical diagnosis, surgical planning, and education. A benchmark that evaluates these dynamics provides a direct path toward clinically relevant applications.

2. **Cellular-level simulations** are central to biology and medicine, as cell migration, proliferation, and interaction underpin processes such as tissue growth, wound healing, and immune response. Accurate modeling at this level enables researchers and students to visualize and understand the driving forces of health and disease, creating opportunities for both discovery and pedagogy.

3. **Subcellular-level simulations** present the most fine-grained view, capturing biochemical and biophysical mechanisms that govern life at its foundation—fusion, apoptosis, signaling cascades. Evaluating generative models at this level is particularly important, as these processes are both visually subtle and mechanistically complex, requiring high fidelity and physical plausibility.

## 2.2 PROMPT SUITE

Both the sampling process of diffusion-based video generation models and the development of expert-driven evaluation rubrics are computationally expensive. To ensure efficiency, we control the number of tasks while maintaining diversity and coverage. The construction follows a two-stage pipeline: *(1) collecting tasks related to microscale simulation from YouTube*; and *(2) expert filtering to retain only scientifically meaningful tasks*. The final suite contains 459 tasks: 238 at the organ level, 189 at the cellular level, and 32 at the subcellular level. The proportion of tasks is consistent with the distribution of levels in the collected videos.

**Collecting and Generating Prompts** We retrieved over 8,000 YouTube videos using topic-specific queries related to organ-level, cellular-level, and subcellular-level simulations. For each video, we collected metadata including titles and descriptions. This information was then provided to GPT-4o, which generated tasks describing the microscale mechanism. Finally, we generated 8,162 tasks. The prompts used to instruct GPT-4o refer to Appendix J.

**Expert Filtering** We filtered the generated tasks based on two criteria: (1) the diversity of the tasks, and (2) the practical relevance of the tasks. For diversity, we asked GPT-4o to classify each task into one of the following categories: Organ-level simulations, Cellular-level simulations, or Subcellular-level simulations. For practical relevance, we invited three biology experts, and each task had to receive agreement from at least two of the three experts. A task was retained in MicroWorldBench only if it satisfied both criteria. Classification prompts are in Appendix K.

## 2.3 RUBRIC CRITERIA

As shown in Figure 2, each MicroWorldBench example includes a task instruction and rubric criteria, drafted by LLMs and refined by experts. These criteria evaluate scientific fidelity, visual quality, and instruction following. Scientific fidelity emphasizes mechanistic accuracy rather than visual realism. An LLM-based grader then scores the output, providing a standardized, interpretable assessment.

Due to limited expert availability and efficiency concerns, we adopt a collaborative approach where LLMs generate initial rubric drafts and experts perform revision and validation. This method not only improves the efficiency of rubric construction but also ensures broader coverage and more comprehensive consideration despite the small number of experts.

**Stage 1: Rubric Drafts Generation** For each task, GPT-5 generates a set of fine-grained criteria: $P = (a_i, d_i, s_i, w_i)_{i=1}^{N}$, where $a_i$ denotes the evaluation dimension, $d_i$ is the description of the $i$-th criterion, $s_i \in +1, -1$ is the polarity indicating whether the point contributes $(+1)$ or deducts $(-1)$, and $w_i \in (0, 1]$ is the weight reflecting its importance (e.g., $w_i = 1.0$ for core scientific requirements, $w_i = 0.5$ for key but secondary requirements, and $w_i = 0.2$ for auxiliary or presentational).

The score for each task is defined as: $S = \sum_{i=1}^{N} s_i \cdot w_i$. To ensure comparability across tasks, we normalize it: $S_{\text{norm}} = \frac{S}{\sum_{i=1}^{N} w_i^+} \times 100$ where $\sum w_i^+$ is the maximum score from positive criteria, ensuring a maximum of 100 and preventing minor positives from offsetting severe scientific errors."

**Stage 2: Expert Revision and Validation** Domain experts refine the LLM-generated rubric through the following actions:

- Deleting or filtering criteria: Experts refine the criteria by modifying or removing $d_i$ that are redundant, irrelevant, or scientifically trivial.

- Adjusting weights: When the weight of certain criteria does not align with the scientific validity of the task, experts modify the corresponding weight $w_i$.

Table 1: Performance comparison of different video generation models on MicroWorldBench. Bold indicates the best performance.

| Model | Average ↑ | Organ-level ↑ | Cellular-level ↑ | Subcellular-level ↑ |
|---|---|---|---|---|
| **Open-Source Video Generation Models** | | | | |
| HunyuanVideo | 23.2 | 23.1 | 23.8 | 19.4 |
| CogVideoX-5B | 43.5 | 39.9 | 47.0 | 38.6 |
| Wan2.1-T2V-1.3B | 49.4 | 45.9 | 51.7 | 52.4 |
| Wan2.2-TI2V-5B | 51.6 | 46.6 | 53.9 | 49.5 |
| Wan2.1-T2V-14B | **54.8** | 55.7 | **54.4** | 52.8 |
| Wan2.2-T2V-A14B | 53.8 | **56.3** | 52.0 | 53.3 |
| **MicroVerse-1.3B (Ours)** | 50.2 | 47.6 | 51.7 | **53.3** |
| **Commercial Video Generation Models** | | | | |
| Sora | 50.7 | 55.9 | 46.1 | 55.0 |
| Veo3 | **77.2** | **77.5** | **76.9** | **78.2** |

Table 2: Performance comparison of different video generation models on MicroWorldBench (dimension-wise scores). Bold indicates the best performance.

| Model | Average ↑ | Scientific Fidelity ↑ | Visual Quality ↑ | Instruction Following ↑ |
|---|---|---|---|---|
| **Open-Source Video Generation Models** | | | | |
| HunyuanVideo | 23.2 | 15.6 | 48.2 | 23.4 |
| CogVideoX-5B | 43.5 | 37.4 | 64.1 | 38.6 |
| Wan2.1-T2V-1.3B | 49.4 | 40.3 | 71.8 | 50.1 |
| Wan2.2-TI2V-5B | 51.6 | 40.7 | 82.7 | 47.0 |
| Wan2.1-T2V-14B | **54.8** | 42.7 | 86.0 | 53.8 |
| Wan2.2-T2V-A14B | 53.8 | 37.8 | **92.8** | **55.4** |
| **MicroVerse-1.3B (Ours)** | 50.2 | **43.0** | 68.5 | 49.3 |
| **Commercial Video Generation Models** | | | | |
| Sora | 50.7 | 35.3 | 96.4 | 37.9 |
| Veo3 | **77.2** | **65.7** | **97.0** | **77.0** |

- Supplementing criteria: If the automatically generated criteria fail to cover essential scientific dimensions, experts can introduce new tuples $(a_j, d_j, s_j, w_j)$.

We invited three experts to participate in the revision and validation process. Each expert first independently reviewed and modified the evaluation criteria, including adjusting weights, removing redundant items, and supplementing any missing dimensions. All modifications were documented with clear rationale to ensure transparency. The proposed changes from all experts were then aggregated, and conflicts were resolved through discussion, majority voting. For more analysis on expert revision and validation, refer to the Appendix C.

## 2.4 EVALUATION RESULTS AND ANALYSIS

**Settings** We evaluated video generation models on microscopic simulation tasks using MicroWorldBench, including open-source models (e.g., Wan2.1 Wan et al. (2025), HunyuanVideo Kong et al. (2024)) and commercial models (e.g., Sora OpenAI (2024), Veo3 Google DeepMind (2025)). Inference was conducted once per model under default settings to ensure fairness and consistent resolution. Rubric evaluation employed LLM-as-a-Judge Zheng et al. (2023), with GPT-5 serving as the Judge. The configurations and sampling details in the Appendix E.

**Overall Results** As shown in Table 1, the performance of different models varies significantly across organ-level, cellular-level, and subcellular-level tasks. Although commercial closed-source models, such as Veo3, substantially outperform open-source models in overall scores, their advantage is mainly confined to the visual quality dimension rather than scientific fidelity.

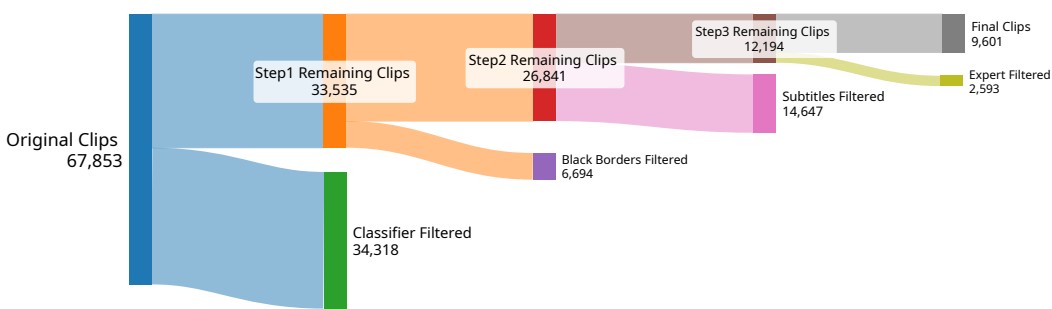

Figure 3: Overview of our data filtering pipeline. Each stage applies specific filters and shows the volume of data removed and retained.

**Visual Quality vs. Scientific Fidelity** Table 2 shows that nearly all models achieve high scores in visual quality (80–97), yet their scientific fidelity lags far behind (most open-source models score only 15–43). This result demonstrates that current models often generate videos that "look right" but fail to strictly adhere to physical and biological laws.

**Performance Differences Across Hierarchical Tasks** Both advanced open-source models (e.g., Wan2.2-T2V-A14B) and top commercial models (Sora, Veo3) exhibit lower performance on cellular and subcellular tasks compared to organ-level simulations. This may be attributed to the higher requirements for physical and biological consistency in these tasks, as well as the scarcity of microscale training data that can capture complex dynamics.

**Scale Effects in Open-Source Models** Within the Wan series, increasing model size from 1.3B to 14B mainly improves visual quality, while scientific fidelity shows little significant growth. This suggests that expanding model parameters alone is not sufficient to solve the core scientific fidelity challenges in microscale simulation.

## 3 MICROVERSE: TOWARD MICROSCALE SIMULATION VIA A EXPERT-VERIFIED DATASET

The results of MicroWorldBench indicate that current models remain limited in their ability to model microscale mechanism governed by physical and biological principles. Most large-scale video datasets—such as InternVid Wang et al. (2023b), UCF101 Soomro et al. (2012), and OpenVid-1M Nan et al. (2024)—primarily consist of natural scenes or human activities, offering little relevance to microscopic processes. To address this challenge, we propose a new microscale simulation models, termed *MicroVerse*, which explicitly incorporate physical grounding and fine-grained biological dynamics. A key prerequisite for developing such models is the availability of domain-specific data that accurately capture microscopic processes with physical fidelity.

### 3.1 DATA CONSTRUCTION: MICROSIM-10K

**Collecting videos from YouTube** We used the official YouTube API to search for videos related to microsimulation and filtered them based on the following criteria: (1) resolution of at least 720p; and (2) licensed under Creative Commons. These requirements ensure that the collected videos are suitable and freely available for training. In total, we obtained 12,848 relevant videos.

**Splitting videos** After obtaining the videos, we segmented them into multiple semantically consistent and short clips. We used OpenCLIP Ilharco et al. (2021) for video segmentation: whenever the similarity between adjacent frames fell below 0.85, a split was made. In total, 67,853 clips were generated. Since not all clips were related to microsimulation, we trained a classifier based on VideoMAE Tong et al. (2022) to filter them. The model achieved an accuracy of over 92%, significantly improving the quality of the dataset. With the help of the classifier, 34,318 clips were filtered out. For details of the clip classification model related to microsimulation, refer to the Appendix L.

**Automatic and expert filtering** To improve the quality and physical consistency of the clips, we first applied OpenCV [1] to detect black borders and used EasyOCR [2] to detect subtitles in order to filter out those affecting semantic representation, retaining 12,194 clips. Experts then reviewed the data, removing meaningless or physically inconsistent clips, resulting in 9,601 clips.

**Generating captions** We leverage a multimodal LLM (GPT-4o) to generate detailed captions. Due to context limits, we uniformly sampled 8 frames per clip as visual input. To minimize hallucinations, we supply the video title and description.

> **Prompt MLLM to Generate Video Caption**
>
> The provided images are sampled from a video clip (8 evenly spaced frames). This clip is taken from a video with the following metadata:
> Video Title: {Video Title}; Video Description: {Video Description}
> Using the visual content of the clip, together with the title and description, please generate a clear, detailed, and accurate description of what is shown. Focus on the subject, explains the scene and actions, and emphasizes visible details, textures, and fine structures.

## 3.2 DATA STATISTICS

### 3.2.1 FUNDAMENTAL ATTRIBUTES

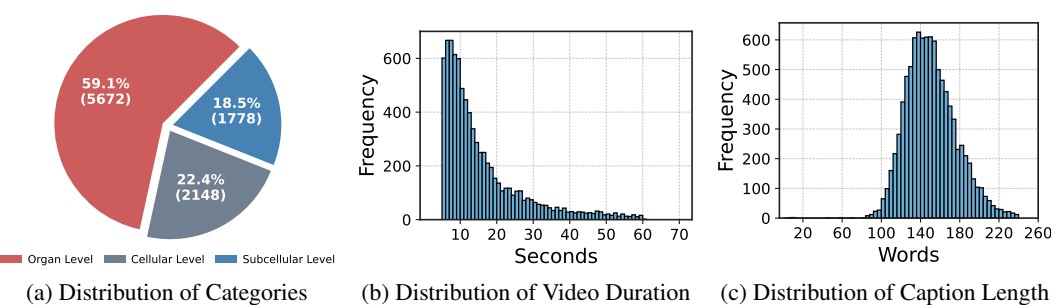

(a) Distribution of Categories    (b) Distribution of Video Duration    (c) Distribution of Caption Length

Figure 4: Distributions of fundamental video attributes in the MicroSim-10K.

MicroSim-10K is the first large-scale dataset dedicated to microscale simulation, comprising 9,601 high-quality video clips. As shown in Figure 4, all clips have a resolution of at least 720p and a duration of 5–60 seconds, ensuring that each captures a complete and coherent microscopic process. The dataset spans diverse biological mechanisms across organ, cellular, and subcellular levels, offering broad coverage of key scenarios. Each clip is paired with a detailed caption generated by a multimodal LLM and validated by experts, with an average length of around 150 words, providing precise semantic alignment for model training.

### 3.2.2 POPULARITY AND RELEVANCE

To capture the educational and communicative value of microscale simulations, MicroSim-10K retains metadata such as views, likes, and comments. As shown in Figure 5, the videos in MicroSim-10K have been widely viewed, with many reaching hundreds of thousands of views, and they have received substantial likes and comments, reflecting strong popularity and broad accessibility across both scientific and public communities.

### 3.2.3 REALISM AND DISTRIBUTION

We compare its distribution with real-world microscopy videos using Fréchet Video Distance (FVD). Using the method described in Section 3.1, we collected 377 real biological videos from YouTube and obtained 643 video clips after preprocessing. As shown in Table 3, the FVD between MicroSim-10K and real biological videos is 123.9. This result indicates that our expert-verified MicroSim-10K

---

[1]https://github.com/opencv/opencv-python
[2]https://github.com/JaidedAI/EasyOCR

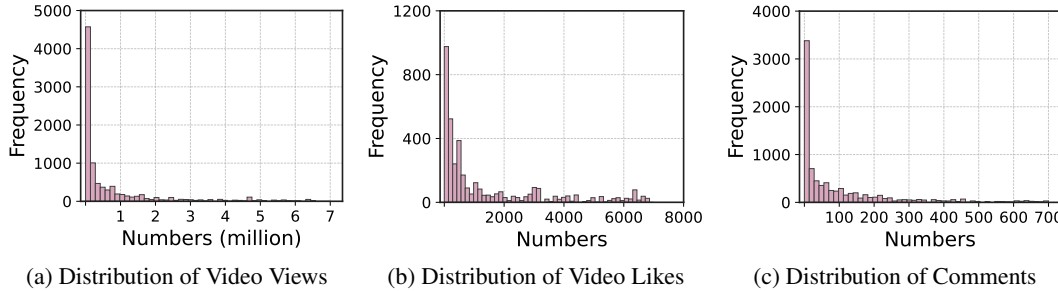

(a) Distribution of Video Views    (b) Distribution of Video Likes    (c) Distribution of Comments

Figure 5: Distributions of video popularity indicators in the MicroSim-10K.

already lies remarkably close to the real microscopy distribution in terms of visual statistics and structural descriptors, effectively bridging the gap between simulated and real experimental data.

Table 3: FVD comparison across models (lower is better). The FVD between MicroSim-10K and real biological videos is 123.9, indicating a close distributional alignment.

| Data | FVD vs. MicroSim-10K ↓ | FVD vs. 643-Real-Biological-Clips ↓ |
|---|---|---|
| MicroSim-10K | 0 | 123.9 |
| 643-Real-Biological-Clips | 123.9 | 0 |
| *Commercial Models* | | |
| Veo3 | 42.6 | 118.1 |
| Sora | 116.9 | 136.3 |
| *Open-Source Models* | | |
| Wan2.1-T2V-1.3B | 83.0 | 158.9 |
| Wan2.2-TI2V-5B | 77.6 | 153.3 |
| Wan2.1-T2V-14B | 53.3 | 137.6 |
| Wan2.2-T2V-A14B | 65.8 | 132.2 |

## 3.3 TRAINING MICROVERSE

For training, we fine-tune the Wan2.1 model. A text prompt $P$ is encoded as a sequence: $P = (p_0, p_1, \ldots, p_m)$, while the target video $V$ is decomposed into $T$ frames. Each frame is mapped into the latent space via a VAE Kingma & Welling (2013) encoder, yielding the sequence: $L = (l_0, l_1, \ldots, l_T)$. The text input $P$ is transformed into embeddings $E$ using CLIP text encoder, and the latent sequence $L$ is processed by a Diffusion Transformer (DiT) Peebles & Xie (2023).

The training objective is to predict the latent representation of the video through a denoising diffusion process. At timestep $t$, the loss function is defined as:

$$\mathcal{L} = \mathbb{E}\left[\|\varepsilon - \varepsilon_\theta(L_t, t, E)\|^2\right],  \tag{1}$$

where $L_t$ is the noisy latent representation at timestep $t$, $\varepsilon$ denotes the injected noise, $\varepsilon_\theta$ is the model's noise prediction, $t$ is the current diffusion timestep, and $E$ is the text embedding.

During fine-tuning, with probability defined by the 10%, the text conditioning is entirely masked, enabling Classifier-Free Guidance (CFG) Ho & Salimans (2022) training. This mixture of unconditional and conditional training improves the generation quality of the model during inference.

## 4 EXPERIMENTS

**Experiment Settings** We train MicroVerse using 8 NVIDIA H200 GPUs, fully fine-tuning all parameters of Wan2.1-T2V-1.3B Wan et al. (2025) with a learning rate of 1e-5 and a batch size of 8. The training process is designed to improve the model's capability to generate microscopic simulation videos conditioned on text prompts. We conducted a comparative with other models on MicroWorldBench. Additional training details are provided in the Appendix D.

**Human Evaluation** To evaluate alignment with human preferences, we conducted a human study comparing MicroVerse with Sora and Veo3. The evaluation included 60 samples across three levels of microsimulation (20 samples per level), all sourced from the 20 most popular microsimulation videos on YouTube. Model outputs were randomly shuffled, and three evaluators independently selected the preferred result based on instruction fidelity and visual clarity, or marked a tie. The final results were reported as preference ratios.

## 4.1 Results of our MicroVerse

**Improvement in Scientific Fidelity** Table 2 shows that MicroVerse achieves a significant improvement in Scientific Fidelity, reaching a score of 43.0 and outperforming all open-source models. This enhancement is attributed to the training on the physics-grounded MicroSim-10K dataset, which enables the model to better adhere to biological and physical laws. Although there is a slight decrease in Visual Quality (68.5) and Instruction Following (49.3), this does not affect our core objective: advancing scientific fidelity.

**Breakthrough in Subcellular-Level Tasks** According to Table 1, on the highly challenging subcellular-level tasks, MicroVerse achieves a score of 53.3, surpassing all open-source models. This demonstrates that our dataset enables MicroVerse to make notable progress on microscale simulation tasks where existing models typically struggle.

## 4.2 Analysis

**Scaling Results** We identify two main limitations in the performance of the 1.3B model: first, the improvement in scientific fidelity is relatively modest when fine-tuning a small-parameter model; second, there is a slight decline in visual quality and instruction-following capabilities. To address these issues, we scale the model parameters to 14B and employ a mixed-domain training strategy, combining MicroSim-10K with an equivalent amount of high-quality general-domain data randomly sampled from OpenVid Nan et al. (2024). As shown in Table 4, this dual scaling of both model capacity and data diversity significantly enhances performance across all dimensions, achieving state-of-the-art results among open-source models. For more ablation studies on dataset filtering, dataset size, and training recipes, please refer to Appendix B.

Table 4: Impact of Data Scale and Mixed-Domain Training on MicroVerse Performance. **Bold** indicates the best performance. FT indicates fine-tuning.

|  | Data Size | Scientific Fidelity ↑ | Visual Quality ↑ | Instruction Following ↑ |
|---|---|---|---|---|
| *Base Model: Wan2.1-1.3B* | | | | |
| Baseline | 0 | 40.3 | 71.8 | 50.1 |
| FT on MicroSim-10K | 9,601 | 43.0 | 68.5 | 49.3 |
| FT on General + MicroSim-10K | 19,202 | **44.1 (+3.8)** | **74.4 (+2.6)** | **53.8 (+3.7)** |
| *Base Model: Wan2.1-14B (Full-Blooded Model)* | | | | |
| Baseline | 0 | 42.7 | 86.0 | 53.8 |
| FT on MicroSim-10K | 9,601 | 45.4 | 82.7 | 51.4 |
| FT on General + MicroSim-10K | 19,202 | **48.3 (+5.6)** | **87.7 (+1.7)** | **56.9 (+3.1)** |

**Human Evaluation Results** Figure 6 shows the results of human evaluation. Compared with Wan2.1-1.3B models, MicroVerse performs excellently in the dimension of Scientific Fidelity. Its outstanding performance in Scientific Fidelity further validates the effectiveness of MicroSim-10K. In addition, the Cohen's Kappa coefficient among the three independent experts was above 0.80, indicating strong interrater agreement and confirming the reliability of the scoring process. More details on Cohen's Kappa coefficient can be found in Appendix G.

**Consistency among Judgers in MicroWorldBench** To ensure that MicroWorldBench's evaluation aligns closely with human judgment across all dimensions, we conducted human preference labeling on a large set of generated videos. Specifically, we computed the consistency of evaluation tasks across different models as well as between the models and humans. Figure 6 shows the consistency relationships among different models and between the models and humans. For more analysis results on evaluation consistency, please refer to Appendix C.

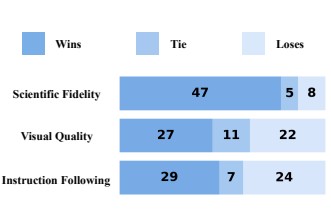 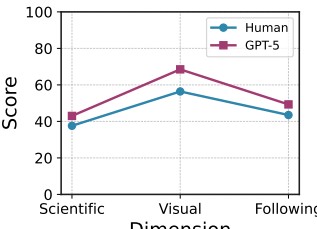 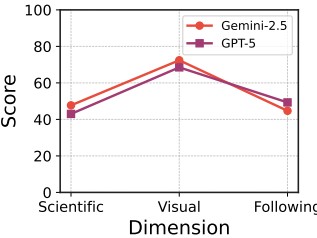

(a) Human Evaluation Result of MicroVerse and Wan2.1-1.3B

(b) Consistency between LLMs and Humans

(c) Consistency across different LLMs

Figure 6: Human Evaluation and Consistency Results.

## 5 RELATED WORK

**World Model** World models LeCun (2022); Bruce et al. (2024); Lu et al. (2024) have garnered significant attention. They simulate dynamic environments by predicting future states and estimating rewards based on current observations and actions. Their ability to model state transitions has been extended to real-world scenarios through joint learning of policies and world models, improving sample efficiency in simulated robotics Seo et al. (2023), real-world robots Wu et al. (2022), clinical decision Yang et al. (2025), and autonomous driving Wang et al. (2023a). For example, some work Du et al. (2023) explores long-horizon video planning by combining vision–language and text-to-video models. Others Luo & Du (2024) focus on linking video models to continuous actions through goal-conditioned exploration. Recent works Lu et al. (2024) also use video generative models to let agents explore environments more effectively. MeWM Yang et al. (2025) applies world modeling to medical image analysis and clinical decision-making.

**Video Generation** Video generation has seen rapid progress in the past two years. The release of Sora OpenAI (2024) has ignited strong research interest in text-to-video generation, leading to breakthroughs in quality, coherence, and controllability Blattmann et al. (2023). Other commercial systems such as Veo3, Kling, HunyuanVideo Kong et al. (2024), and Hailuo HailuoAI (2024) have achieved impressive performance and are widely applied in video production, advertising, and education. With the technology maturing, domain-specific models are emerging to address specialized needs. For instance, MedGen Wang et al. (2025) generates accurate, high-quality medical videos for health education, while AniSora Jiang et al. (2025) focuses on producing detailed and stylistically rich animated content. Despite these advances, the use of video generation for microscale simulation remains largely unexplored.

**Rubric Evaluation** Rubric-based evaluation has become a standard approach for assessing LLMs on open-ended tasks, offering task-specific and interpretable criteria that improve grading consistency. HealthBench Arora et al. (2025) scales this paradigm to 5,000 multi-turn conversations with 48k clinician-authored rubrics covering accuracy, safety, and communication. Building on this, Baichuan-M2 Team (2025) dynamically generates case-specific rubrics as verifiable reward signals for reinforcement learning, enabling adaptive and context-aware supervision. Rubrics as Rewards (RaR) Gunjal et al. (2025) further formalizes rubric-based RL and shows significant gains over Likert-style scoring. These efforts highlight rubric-guided evaluation and training as a promising methodology for developing reliable, aligned, and LLMs.

## 6 CONCLUSIONS

Video generation excel at natural and human-centered macroscopic scenes but fail to capture faithful microscale dynamics. This work introduces MicroWorldBench, the first rubric-based benchmark for microscale video generation with 459 expert-curated tasks and well-defined rubric criteria. In addition, we build MicroSim-10K and develop MicroVerse which demonstrate remarkable performance on microscale simulation tasks. By integrating physical constraints and expert supervision, MicroVerse not only improves visual fidelity but also advances toward biologically meaningful dynamics, enabling applications in biomedical research, education, and interactive scientific visualization.

## LIMITATION

Our work aims to explore the potential of educational microscale simulations of biological mechanisms, rather than the reproduction of results observed in wet lab experiments. However, our current approach does not explicitly incorporate the underlying physical laws that govern biomedical microscale dynamics, such as fluid mechanics in blood flow, diffusion–reaction equations in molecular transport, or biomechanical constraints in cellular processes. This limitation restricts the applicability of the model in scenarios that require high-precision scientific simulation and prediction.

## ETHICS STATEMENT

All data are publicly available, compliant with YouTube's terms, and we exclude personal/sensitive content. Captions were auto-generated (MLLMs) and manually verified to remove inappropriate/identifiable material. The dataset is intended solely and strictly for research purposes and should not be used for non-research settings. We do not own the copyright of these data and will only publicly release the URLs linked to the data instead of the raw data.

## ACKNOWLEDGEMENTS

This work was supported by Major Frontier Exploration Program (Grant No. C10120250085) from the Shenzhen Medical Academy of Research and Translation (SMART), Shenzhen Medical Research Fund (B2503005), the Shenzhen Science and Technology Program (JCYJ20220818103001002), NSFC grant 72495131, Shenzhen Doctoral Startup Funding (RCBS20221008093330065), Tianyuan Fund for Mathematics of National Natural Science Foundation of China (NSFC) (12326608), Shenzhen Science and Technology Program (Shenzhen Key Laboratory Grant No. ZDSYS20230626091302006), the 1+1+1 CUHK-CUHK(SZ)-GDSTC Joint Collaboration Fund, Guangdong Provincial Key Laboratory of Mathematical Foundations for Artificial Intelligence (2023B1212010001), the International Science and Technology Cooperation Center, Ministry of Science and Technology of China (under grant 2024YFE0203000), and Shenzhen Stability Science Program 2023.

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

# A DATA FILTERING PIPELINE

As illustrated in Figure 7, our data construction process is designed to ensure high fidelity and semantic richness. We start with an initial collection of 128K microsimulation videos. To guarantee data quality, we implement a rigorous cleaning protocol that filters out non-microsimulation samples and eliminates temporal inconsistencies. Furthermore, we utilize OpenCV to crop black borders and employ EasyOCR to detect and remove hard-coded subtitles, thereby reducing visual noise. The preprocessed videos undergo a human-in-the-loop verification stage, where experts assess the content for meaningfulness and consistency. In the final stage, we leverage Multimodal Large Language Models (MLLMs) to automatically generate descriptive captions, resulting in a high-quality dataset suitable for downstream representation learning tasks.

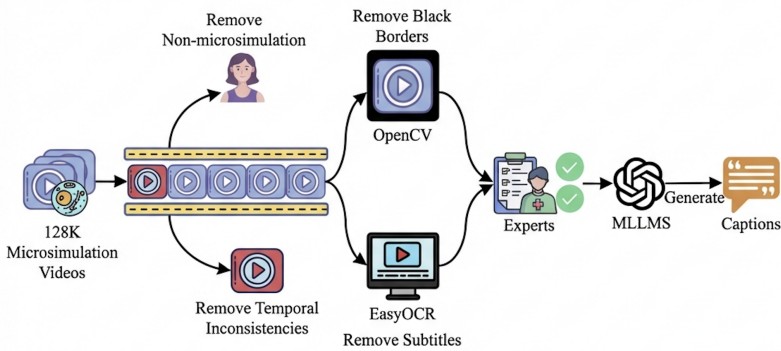

Figure 7: Overview of the data construction process.

To ensure absolute data purity and avoid potential data leakage, we implemented a rigorous three-level deduplication pipeline. As shown in Table 5, we found zero overlap at high similarity thresholds across source, text, and vision levels.

Table 5: Three-level deduplication pipeline results.

| Deduplication Type | Method | Metric | Result |
|---|---|---|---|
| Source-level | Video ID matching | # overlapping IDs | 0 |
| Text-level | Caption embedding | # pairs $> 0.90$ | 0 |
| Vision-level | Frame embedding | # pairs $> 0.95$ | 0 |

Furthermore, we lowered the text-level and vision-level thresholds to 0.60 to help remove potentially overlapping tasks and ensure absolute data purity. Table 6 shows the task distribution after deduplication.

Table 6: Task distribution after deduplication.

| Biological Level | Original | Post-Deduplication | Removed |
|---|---|---|---|
| Organ-level | 238 | 233 | -5 |
| Cellular-level | 189 | 182 | -7 |
| Subcellular-level | 32 | 30 | -2 |
| **Total** | **459** | **445** | **-14** |

We re-evaluated the model performance on the clean test set (445 tasks), as shown in Table 7. The performance scores remained highly stable, with only negligible variations (decimal-level drops), therefore we ultimately adopted the 459 tasks.

Regarding private models like Veo3, we acknowledge they may have seen YouTube data. However, MicroWorldBench evaluates scientific fidelity, not just generalization. Even if a model has seen similar data, generating a simulation that strictly adheres to our expert-weighted physical rubrics remains a valid measure of capability. We plan to expand the benchmark with specialized microscopy databases in the future to broaden the domain distribution.

Table 7: Performance comparison on original vs. clean test set.

| Test Set | Average | Scientific Fidelity | Visual Quality | Instruction Following |
|---|---|---|---|---|
| Original (459 tasks) | 50.2 | 43.0 | 68.5 | 49.3 |
| Clean (445 tasks) | 50.0 | 42.7 | 68.5 | 49.1 |

## B  ABLATION STUDY

In this section, we present a series of ablation studies to evaluate the impact of different components and hyperparameters on the model's performance.

**Ablation Study on dataset filtering.**  Filtered data (MicroSim-10K) yields more balanced and reliable improvements than using raw, uncleaned data, boosting scientific fidelity while avoiding major drops in visual quality and instruction following. Table 8 summarizes the results.

Table 8: Ablation study on dataset filtering.

| Variant | Data Size | Sci. Fidelity | Vis. Quality | Inst. Following |
|---|---|---|---|---|
| Baseline Model (Wan2.1-1.3B) | 0 | 40.3 | 71.8 | 50.1 |
| FT on raw (uncleaned) data | 34,318 | 42.1 | 67.4 | 44.5 |
| FT on MicroSim-10K | 9,601 | 43.0 | 68.5 | 49.3 |

**Ablation Study on dataset size.**  Increasing the size of high-quality training data yields steady gains in scientific fidelity and instruction following with only minor visual-quality tradeoffs. Table 9 summarizes the results.

Table 9: Ablation study on dataset size.

| Variant | Data Size | Sci. Fidelity | Vis. Quality | Inst. Following |
|---|---|---|---|---|
| Baseline Model (Wan2.1-1.3B) | 0 | 40.3 | 71.8 | 50.1 |
| FT on MicroSim-10K (50% sample) | 4,800 | 41.9 | 69.1 | 48.4 |
| FT on MicroSim-10K | 9,601 | 43.0 | 68.5 | 49.3 |

**Ablation Study on training recipes (CFG rate).**  Excessive CFG sharply degrades scientific fidelity and visual quality. Table 10 summarizes the results.

**Ablation Study on training recipes (training steps).**  As shown in the training curves, the model converges rapidly and stabilizes around 5,000 steps. Therefore, we adopt this number of training steps as the default setting for reporting efficiency.

**Ablation Study on training recipes (number of frames).**  We selected 81 frames as a balanced trade-off:

- It provides sufficient temporal duration to capture complete distinct biological events (e.g., cell division) while maintaining high temporal coherence and manageable computational costs.
- Additionally, 81 frames is a native, optimal temporal window for the Wan2.1 architecture Wan et al. (2025).

## C  ANALYSIS OF THE PROCESS OF EXPERT REVISION AND VALIDATION

All experts involved in our evaluation hold doctoral degrees (Ph.D.) and possess extensive research experience in cellular and molecular biology. Table 11 provides the background information for each expert.

Table 10: Ablation study on CFG rate.

| Variant | Data Size | Sci. Fidelity | Vis. Quality | Inst. Following |
|---|---|---|---|---|
| Baseline Model (Wan2.1-1.3B) | 0 | 40.3 | 71.8 | 50.1 |
| FT on MicroSim-10K (CFG=5%) | 9,601 | 41.5 | 70.2 | 49.8 |
| FT on MicroSim-10K (CFG=10%) | 9,601 | 43.0 | 68.5 | 49.3 |
| FT on MicroSim-10K (CFG=15%) | 9,601 | 38.2 | 63.1 | 51.4 |

Table 11: Background information of the experts involved in the evaluation.

| Expert ID | Field of Expertise | Years of Experience |
|---|---|---|
| 1 | Cell Biology/Genetics | 12 years |
| 2 | Biochemistry / Metabolic Pathway Modeling | 10 years |
| 3 | Biochemistry / Metabolic Pathway Modeling | 8 years |

We analyzed the frequency with which three experts employed the four types of rubric operations when handling different tasks. As shown in Figure 8, all experts tended to favor Adjust Weights, while Supplement was used relatively infrequently. Follow-up interviews with the three experts revealed that the Supplement operation is more cumbersome, as it requires identifying additional evaluation criteria beyond those automatically generated by the LLM, which can introduce extra burden.

As detailed in Table 12, experts performed three specific types of interventions to ensure biological and physical processes were captured: (1) **Filtering** scientifically trivial criteria; (2) **Adjusting weights** (e.g., setting $w_i = 1.0$) to prioritize underlying scientific mechanisms over superficial visual quality; and (3) **Supplementing** missing physical constraints. As shown in Table 12, only about 7.6% of the 459 tasks required experts to add missing scientific criteria. This indicates that GPT-5's initial rubric drafts already captured the essential mechanisms, while experts played a crucial role in refining the final outputs.

Furthermore, Table 13 shows that experts intervened more frequently in subcellular-level tasks to enforce strict physical constraints where GPT-5 is less reliable, ensuring scientific accuracy.

Furthermore, we conducted a correlation analysis between human experts and LLMs to validate the reliability of automated evaluation. Three domain experts independently evaluated a subset of videos, and we quantified the scoring agreement between a human-only panel and a panel including GPT-5. As shown in Table 14, the inclusion of GPT-5 maintains or even slightly improves the inter-rater agreement (Fleiss' Kappa).

We also report the agreement across different biological scales in Table 15. Overall, GPT-5 shows agreement with expert scoring at a level comparable to human evaluators, and in some cases its consistency is even slightly higher. This suggests that GPT-5 is a reliable evaluator rather than a source of additional variance.

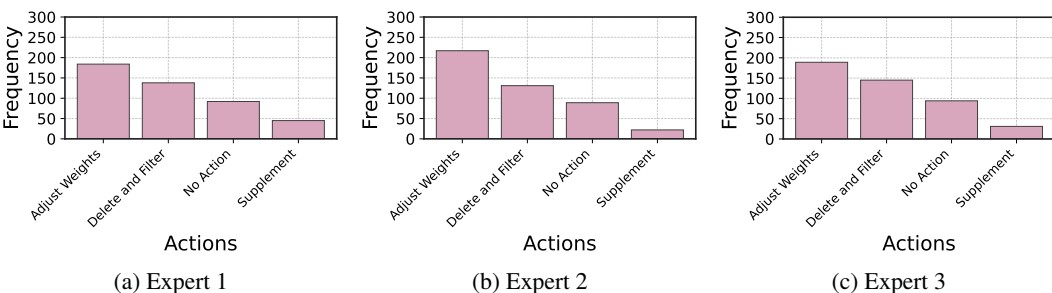

(a) Expert 1      (b) Expert 2      (c) Expert 3

Figure 8: Actions Log of Three Independent Experts.

Table 12: Detailed expert actions during the rubric revision process.

| Action | Expert 1 | Expert 2 | Expert 3 |
|---|---|---|---|
| Adjusting weights | 180 | 221 | 189 |
| Filtering | 137 | 128 | 146 |
| No Action | 94 | 82 | 95 |
| Supplementing | 48 | 28 | 29 |

Table 13: Expert intervention frequency across different biological scales.

| Biological Level | Number of Tasks | Total Expert Actions | Avg. Actions per Task |
|---|---|---|---|
| Organ-level | 238 | 361 | 1.52 |
| Cellular-level | 189 | 567 | 3.00 |
| Subcellular-level | 32 | 176 | 5.50 |

# D    TRAINING SETTINGS ON MICROVERSE

We implemented our training pipeline utilizing a high-performance computational node equipped with 8 NVIDIA H200 GPUs. To ensure the model fully captures the intricate dynamics and visual nuances specific to the microscopic domain, we adopted a full parameter fine-tuning strategy on the pre-trained Wan2.1-T2V-1.3B model.

We utilized a global batch size of 8 and set the learning rate to $1 \times 10^{-5}$ with a consistent weight decay of 0.01 to prevent overfitting. To maximize computational efficiency and memory utilization without compromising numerical precision, we employed bfloat16 (bf16) mixed-precision training. Furthermore, full gradient checkpointing was enabled to significantly reduce the memory footprint during the backpropagation phase.

For the visual data configuration, the model was trained to generate high-fidelity video sequences with a resolution of $480 \times 832$ pixels and a temporal duration of 81 frames. A training ControlNet/Classifier-Free Guidance (CFG) rate of 0.1 was applied to randomly drop text conditioning, thereby reinforcing the model's capability for unconditional generation and improving overall robustness. Detailed training configurations are enumerated in Table 16.

# E    INFERENCE SETTINGS ON MICROVERSE

To ensure a rigorous and unbiased assessment of generation quality, we standardized the inference protocol across all evaluated models. The inference configuration strictly mirrors the training resolution settings ($480 \times 832$, 81 frames) to avoid potential domain shifts during the generation phase.

We employed a standard sampling strategy with 50 inference steps, striking an optimal balance between generation latency and visual fidelity. A guidance scale of 5.0 was selected to ensure strong adherence to the text prompts while maintaining natural visual diversity and avoiding artifacts associated with excessive guidance.

Crucially, to uphold the integrity of our evaluation benchmark, we enforced a strict single-shot inference policy. For each test prompt, only a single video sample was generated using a fixed random seed. This approach eliminates the possibility of cherry-picking—selecting the best output from multiple attempts—thereby providing an honest reflection of the model's stability and generalized performance. The comprehensive parameter settings for inference are listed in Table 17.

Table 14: Scoring agreement between human experts and LLMs.

| Evaluation Group | Agreement Score |
|---|---|
| Human-only (Fleiss' Kappa) | 0.733 |
| Human + GPT-5 (Fleiss' Kappa) | 0.771 |
| GPT-5 vs Gemini-2.5-pro (Spearman) | 0.835 |
| GPT-5 vs GPT-5 (Spearman) | 0.912 |

Table 15: Agreement across biological scales.

| Level | Agreement (Human-only) | Agreement (Human + GPT-5) |
|---|---|---|
| Organ-level | 0.728 | 0.742 |
| Cellular-level | 0.740 | 0.739 |
| Subcellular-level | 0.691 | 0.727 |

Table 16: Detailed hyperparameters used during the training phase of MicroVerse.

| Parameter | Value |
|---|---|
| `--train_batch_size` | 8 |
| `--max_train_steps` | 5000 |
| `--learning_rate` | 1e-5 |
| `--mixed_precision` | bf16 |
| `--training_cfg_rate` | 0.1 |
| `--num_height` | 480 |
| `--num_width` | 832 |
| `--num_frames` | 81 |
| `--weight_decay` | 0.01 |
| `--dit_precision` | fp32 |
| `--enable_gradient_checkpointing_type` | full |

Table 17: Standardized inference parameter settings for model evaluation.

| Parameter | Value |
|---|---|
| `--height` | 480 |
| `--width` | 832 |
| `--num_frames` | 81 |
| `--guidance_scale` | 5.0 |
| `--num_inference_steps` | 50 |

## F PROMPT GPT-5 TO GENERATE RUBRIC CRITERIA

**Prompt GPT-5 to Generate Rubric Criteria**

**Task:**
You are a biology expert. Your task is to **design a set of rubrics** to evaluate the completion of a given task based on the provided **Prompt**.
The rubric should consist of multiple **triplets** in the form:

$$\{a_i, d_i, s_i, w_i\}$$

- $a_i$: the evaluation aspect, restricted to one of the following three categories:
  - Scientific Fidelity: Accurate representation of organs, cells, and subcellular structures in scale, morphology, and spatial relationships, with dynamic processes consistent with biological and physical laws.
  - Visual Quality: Emphasis on clarity, detail, and aesthetics, including model precision, rendering, lighting, and color balance.
  - Instruction Following: Generated videos strictly follow the prompt description.
- $d_i$: description of the $i$-th evaluation criterion.
- $s_i$: polarity of the criterion, either $+1$ (contributes positively) or $-1$ (deducts points); leave this field empty.
- $w_i$: weight of importance in the range $(0, 1]$:
  - $1.0 \rightarrow$ core scientific requirements
  - $0.5 \rightarrow$ important but secondary requirements
  - $0.2 \rightarrow$ auxiliary or presentational requirements

**Output example:**
```
{
  "a1":  "Scientific Fidelity",
  "d1":  "Key cell structures are clearly defined and proportionally
    accurate",
  "s1":  "+1",
  "w1":  "1.0"
}
```

**Constraint:**
1. Do not give any explanation, output directly.
2. Please describe the evaluation criterion ($d_i$) in as much detail as possible.
3. Directly describe the key rubrics in the evaluation criterion, and do not use words such as 'whether'.
4. Only English output is allowed.

**Given prompt:**
{prompt}

## G INTER-RATER RELIABILITY AMONG HUMAN EVALUATORS

We used Cohen's Kappa coefficient to measure agreement among the three experts. Table 18 indicate strong agreement among the experts, confirming the reliability of the scoring process.

Table 18: Comparison of Cohen's Kappa values between experts.

| Comparison Pair | Cohen's Kappa |
|---|---|
| Expert1 vs. Expert2 | 0.87 |
| Expert1 vs. Expert3 | 0.83 |
| Expert2 vs. Expert3 | 0.81 |

## H  A RUBRIC EXAMPLE FROM MICROWORLDBENCH

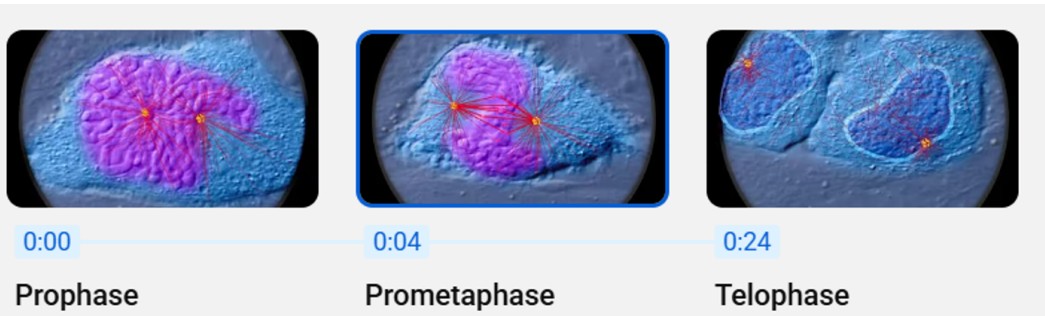

Figure 9: Example of a cell mitosis simulation video frame, taken from an excellent example video on YouTube.

Table 19: Rubric Example for Cell Mitosis Evaluation

| Dimension | Criteria | Weight |
|---|---|---|
| **Scientific Fidelity** | The sequence of stages, segregation patterns, and ploidy changes in mitosis and meiosis are accurately represented; mitosis produces two genetically identical diploid daughter cells, whereas meiosis involves two successive divisions resulting in four genetically diverse haploid gametes. | + 1.0 |
| | The alignment of chromosomes at the metaphase plate and their segregation from metaphase to anaphase occur correctly; sister chromatids are distinctly differentiated from homologous chromosomes; the attachment of kinetochores to spindle microtubules and the direction of their tension conform to biological principles. | + 1.0 |
| | Structural details and dynamic coordination between the spindle apparatus and the centrosome (centriole) are accurate; spindle pole positioning, microtubule polarity, and force distribution are appropriate; the relationship between the microtubule-organizing center and cell polarity is correctly established. | + 1.0 |
| | The timing and mechanisms of DNA replication and genetic recombination are correctly presented; DNA replication occurs during the pre-mitotic S phase, homologous pairing and crossing over take place in prophase I of meiosis, and no DNA replication occurs between meiosis I and II. | + 1.0 |
| **Visual Quality** | The image demonstrates high clarity and fine presentation of microstructural details, with sharp edges of subcellular structures, well-defined layer separation, and absence of wax artifact noise. | + 0.5 |
| | The animation exhibits coherent motion with stable temporal rhythm, smooth phase transitions, and natural movement trajectories, without any stuttering or tearing. | + 0.5 |
| | The 3D modeling and material texture are credible, with consistent form proportions and scale hierarchy; the textures are detailed, and surface microstructures are discernible. | + 0.2 |
| | Coordination of lighting, shadows, and depth of field; controlled volumetric scattering and highlights without excess; clear subject contours with well-defined micro-scale detailing. | + 0.2 |
| **Instruction Following** | Accurately present the key stages of mitosis in a single somatic cell, ensuring a clear transition and coherent progression between meiotic divisions I and II in gonadal germ cells. | + 0.5 |
| | Accurate reproduction of subcellular elements and dynamics: chromosome separation following metaphase plate alignment, coordinated movement of spindle fibers and centrioles, and continuous changes and details of the cell membrane/cytokinesis. | + 0.5 |
| | Accurately describe genetic outcomes and differences: mitosis produces two genetically identical diploid daughter cells, while meiosis results in four genetically diverse haploid gametes, highlighting the mechanistic distinctions. | + 0.2 |
| | Compliance with technical specifications and viewing angle requirements: within an approximate total duration of 5 seconds, information is organized clearly; microscopic close-up focuses on the single-cell subject; the camera remains stable, transitions are clear, and the subject is unobstructed. | + 0.2 |
| | Presence of fundamental conceptual and procedural errors: confusion between mitosis and meiosis, incorrect sequencing of stages, inaccurate ploidy descriptions, omission of the two meiotic divisions, or failure to represent the single-cell focus. | - 0.5 |

# I EXAMPLE OF REAL BIOLOGICAL VIDEO CLIPS

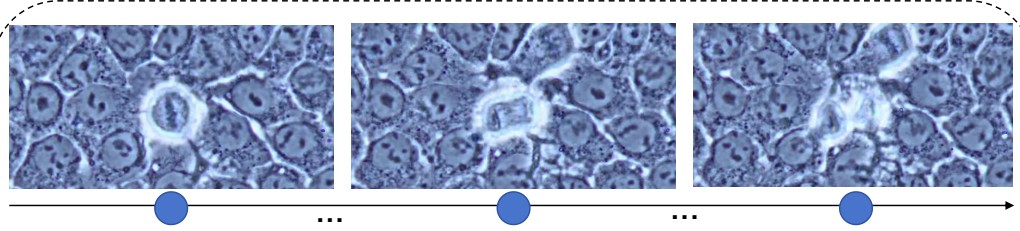

**Caption**: A high-magnification microscopic view reveals a tightly packed layer of rounded, polygonal cells with finely granular cytoplasm and dark, well-defined nuclei, their thin bright borders forming a mosaic-like tissue texture. At the center, one prominent cell is caught in the midst of mitosis, distinguished by its luminous, ring-shaped outline and a nucleus that appears elongated or partially split as condensed chromatin masses align and separate—visual cues of an active division stage. Surrounding cells remain in interphase, showing intact round nuclei with small dark spots suggestive of nucleoli, providing contrast to the dynamic structural rearrangements within the dividing cell. The overall scene captures the subtle shifts in texture, contrast, and intracellular organization characteristic of living cells undergoing cell division.

Figure 10: Example of real biological video clips.

# J GENERATE PROMPT FROM THE VIDEO TITLE AND DESCRIPTION

**Generate Prompt from the Video Title and Description.**

Your task: Based on the "video title" and "video description" I provide, craft an extremely detailed, professional, and keyword-rich English prompt specifically for generating breathtaking microscopic-world videos. Do not give any explanation, output directly. Don't use bullet points. Write it as a single, complete paragraph.

Generate a complete, ready-to-use video-generation prompt. This prompt must include all of the following sections:

1. Main Subject: Clearly describe the central object within the microscopic scene.

2. Scene & Action: Describe what is happening.

3. Details & Textures: Emphasize the details that should be visible.

The video title is:
{video_title}
The video description is:
{video_description}

# K PROMPT LLM CLASSIFIES BASED ON TASK DESCRIPTIONS

**Prompt LLM Classifies Based on Task Descriptions.**

Your task: You are an expert in scientific video classification. Given a task description, classify it into one of the following categories for diversity:

1. Organ-level simulations – tasks focusing on the behavior, dynamics, or interactions at the scale of whole organs or organ systems.

2. Cellular-level simulations – tasks focusing on the behaviors and interactions of single cells or collections of cells, such as cell division, cell fusion, cell migration, or cell signaling.

3. Subcellular-level simulations – tasks focusing on molecular, genetic, or biochemical processes within cells, such as protein folding, gene regulation, or intracellular signaling.

The task description is:
{Task_Description}
Please output only the most appropriate category label based on the task description provided.

## L  VIDEOMAE-BASED MICROSIMULATION CLASSIFIER

To filter out video clips related to microsimulation, we trained a classifier using 2,580 manually annotated samples based on the VideoMAE model. The training was implemented within the Transformers, with a learning rate of 5e-5 and a total of 10 epochs, enabling the model to effectively capture video features and achieve accurate classification. Finally, our classifier achieved an accuracy of 92% on the test set.

Table 20: Dataset statistics for microsimulation classification.

| Category | Total | Train (80%) | Test (20%) |
|---|---|---|---|
| Microsimulation-related | 1107 | 885 | 222 |
| Non-microsimulation | 1473 | 1178 | 295 |
| **Total** | 2580 | 2063 | 517 |

