# OpenReview forum: "MicroVerse: A Preliminary Exploration Toward a Micro-World Simulation"
_ICLR.cc/2026/Conference — ICLR 2026 Poster_

### Official Review · Reviewer_9ESj · 2025-10-29

**Soundness:** 3
**Presentation:** 3
**Contribution:** 2
**Rating:** 6
**Confidence:** 4

**Summary:**

The paper introduces a video generation benchmark, MicroWorldBench, and a video generation model, MicroVerse, to explore how well video generative models perform on microscale simulation tasks. MicroWorldBench is a rubric-based benchmark with 459 tasks spanning organ, cellular, and subcellular levels. The authors also construct MicroSim-10K, a dataset of 9,601 expert-verified microscale simulation videos, and fine-tune Wan2.1 on MicroSim-10K to obtain the model MicroVerse. MicroVerse shows some improvement in scientific fidelity over the base model, though with decreased visual quality and instruction-following capabilities.

**Strengths:**

- Novel application domain. Applying video generation to microscale simulation is unexplored and potentially impactful for education and biomedical research.

- Comprehensive benchmark design. The rubric-based evaluation with expert-curated criteria across three hierarchical levels (organ/cellular/subcellular) is well-motivated and systematic.

- Dataset construction effort. Building MicroSim-10K with multiple filtering stages and expert verification demonstrates thoroughness in data curation.

**Weaknesses:**

- Single-domain dataset that may not represent the full distribution and may have data contamination issues. Both the training set (MicroSim-10K) and test set (MicroWorldBench) are built entirely from YouTube videos, which may not reflect the full range of scientific simulation requirements. The authors have not reported how they deduplicate to ensure the test set is fully separate from the training set. Additionally, there is a concern that private models (like Veo3) may have already been trained on all YouTube videos.

- Evaluation metrics may not capture the biological/physical processes. For the test set, MicroWorldBench, the rubrics were first generated by GPT-5 and then verified by humans. While these human-in-the-loop metrics might be useful, GPT-5 is unlikely to capture the biological/physical processes in the video, and thus important metrics related to these are likely missing.

- Missing ablations. No ablation studies on how dataset filtering stages, dataset size, or training recipes (CFG rate, steps, frames) affect the final performance.

**Questions:**

See weaknesses.

Typos: Figure 1 "Mircoscale" -> "Microscale"; Figure 3 "Boraders" -> "Borders".

---

> ### Author Response · Authors · 2025-11-24
> **Reply to Reviewer 9ESj, Part 1**
>
> We sincerely appreciate your valuable feedback. Below, we address each concern.
>
> > **Weakness 1: Single-domain dataset that may not represent the full distribution and may have data contamination issues. Both the training set (MicroSim-10K) and test set (MicroWorldBench) are built entirely from YouTube videos, which may not reflect the full range of scientific simulation requirements. The authors have not reported how they deduplicate to ensure the test set is fully separate from the training set. Additionally, there is a concern that private models (like Veo3) may have already been trained on all YouTube videos.**
>
> **A1**: We appreciate your scrutiny regarding data distribution and contamination. We addressed this through a strict deduplication protocol and re-evaluation.
>
> We implemented a rigorous three-level deduplication pipeline. As shown in table below, we found zero overlap at high similarity thresholds.
>
> | Deduplication Type | Method | Metric | Result |
> | :--- | :--- | :--- | :---: |
> | **Source-level** | Video ID matching | \# overlapping IDs | 0 |
> | **Text-level** | Caption embedding | \# pairs > 0.90 | 0 |
> | **Vision-level** | Frame embedding | \# pairs > 0.95 | 0 |
>
> Furthermore, we lowered the text-level and vision-level threshold to 0.60 to help remove potentially overlapping tasks and ensure absolute data purity. The table below shows that the task distribution after deduplication.
>
> | Biological Level | Original | **Post-Deduplication** | Removed |
> | :--- | :---: | :---: | :---: |
> | Organ-level | 238 | **233** | -5 |
> | Cellular-level | 189 | **182** | -7 |
> | Subcellular-level | 32 | **30** | -2 |
> | **Total** | **459** | **445** | **-14** |
>
> We re-evaluated MicroVerse on the clean test set (445 tasks), as shown in table below. The performance scores remained highly stable, with only negligible variations (decimal-level drops), confirming that our reported performance is not a result of memorization or data leakage.
>
> | Test Set | **Average** | **Scientific Fidelity** | **Visual Quality** | **Instruction Following** |
> | :--- | :---: | :---: | :---: | :---: |
> | Original (459 tasks) | 50.2 | 43.0 | 68.5 | 49.3 |
> | **Clean (445 tasks)** | **50.0** | **42.7** | **68.5** | **49.1** |
>
> Regarding private models like Veo3, we acknowledge they may have seen YouTube data. However, **MicroWorldBench evaluates scientific fidelity, not just generalization**. Even if a model has seen similar data, generating a simulation that strictly adheres to our expert-weighted physical rubrics remains a valid measure of capability.
>
> We plan to expand the benchmark with specialized microscopy databases in the future to broaden the domain distribution.
>
> > **Weakness 2: Evaluation metrics may not capture the biological/physical processes. For the test set, MicroWorldBench, the rubrics were first generated by GPT-5 and then verified by humans. While these human-in-the-loop metrics might be useful, GPT-5 is unlikely to capture the biological/physical processes in the video, and thus important metrics related to these are likely missing.**
>
> **A2**: We clarify that **MicroWorldBench is not merely "verified" by humans, but actively "revised and co-constructed" by them**.
>
> Our human-in-the-loop pipeline (*Section 2.3*) was specifically designed to address the limitation you mentioned. As detailed in **Appendix D**, experts performed three specific types of interventions to ensure biological/physical processes were captured:
> 1. **Filtering**: Experts removed scientifically trivial criteria.
> 2. **Adjusting weights**: Experts adjusted weights (e.g., setting $w_i=1.0$) to prioritize underlying scientific mechanisms over superficial visual quality.
> 3. **Supplementing**: Experts added missing physical constraints.
>
> As shown in **Figure 8 (Appendix D) and the table below**, only **about 7.6% of the 459 tasks** required experts to add missing scientific criteria. This indicates that GPT-5’s initial rubric drafts already captured the essential mechanisms depicted in the videos. Although experts played an important role in refining the final outputs, the foundational conceptual knowledge provided by GPT-5 remains the key factor in identifying the underlying biological and physical processes.
>
> |Action|Expert 1|Expert 2|Expert 3|
> |:-|:-:|:-:|:-:|
> |Adjusting weights|180|221|189|
> |Filtering|137|128|146|
> |No Action|94|82|95|
> |Supplementing|48|28|29|
>
> As shown in the table below, this trend indicates that **experts intervened more frequently in subcellular-level tasks to enforce strict physical constraints where GPT-5 is less reliable, ensuring scientific accuracy**.
>
> | Biological Level | Number of Tasks | Total Expert Actions (Adjust/Filter/Supplement) | **Avg. Actions per Task** |
> | :--- | :---: | :---: | :---: |
> | Organ-level | 238 | 361 | **1.52** |
> | Cellular-level | 189 | 567 | **3.00** |
> | Subcellular-level | 32 | 176 | **5.50** |

---

> ### Author Response · Authors · 2025-11-24
> **Reply to Reviewer 9ESj, Part 2**
>
> > **Weakness 3: Missing ablations. No ablation studies on how dataset filtering stages, dataset size, or training recipes (CFG rate, steps, frames) affect the final performance.**
>
> **A3**:
>
> **Exp. 1. Ablation Study on dataset filtering.**
>
> Filtered data (MicroSim-10K) yields more balanced and reliable improvements than using raw, uncleaned data, boosting scientific fidelity while avoiding major drops in visual quality and instruction following. Table summarizes the results.
>
> |Variant|Data Size|Scientific Fidelity|Visual Quality|Instruction Following|
> |:-|:-|:-:|:-:|:-:|
> |Baseline Model (Wan2.1-1.3B)|0|40.3|71.8|50.1|
> |FT on raw (uncleaned) data|34,318|42.1|67.4|44.5|
> |FT on MicroSim-10K|9,601|43.0|68.5|49.3|
>
> **Exp. 2. Ablation Study on dataset size.**
>
> Increasing the size of high-quality training data yields steady gains in scientific fidelity and instruction following with only minor visual-quality tradeoffs. Table summarizes the results.
>
> |Variant|Data Size|Scientific Fidelity|Visual Quality|Instruction Following|
> |:-|:-|:-:|:-:|:-:|
> |Baseline Model (Wan2.1-1.3B)|0|40.3|71.8|50.1|
> |FT on MicroSim-10K (Randomly sample half of the data from MicroSim-10K)|4,800|41.9|69.1|48.4|
> |FT on MicroSim-10K|9,601|43.0|68.5|49.3|
>
> **Exp. 3. Ablation Study on training recipes (CFG rate).**
>
> Excessive CFG sharply degrades scientific fidelity and visual quality. Table summarizes the results.
>
> |Variant|Data Size|Scientific Fidelity|Visual Quality|Instruction Following|
> |:-|:-|:-:|:-:|:-:|
> |Baseline Model (Wan2.1-1.3B)|0|40.3|71.8|50.1|
> |FT on MicroSim-10K (CFG=5%)|9,601|41.5|70.2|49.8|
> |FT on MicroSim-10K (CFG=10%)|9,601|43.0|68.5|49.3|
> |FT on MicroSim-10K (CFG=15%)|9,601|38.2|63.1|51.4|
>
> **Exp. 4. Ablation Study on training recipes (training steps).**
>
> As shown in the newly added **Figure 9 in the Appendix E**, the model converges rapidly and stabilizes around 5,000 steps. Therefore, we adopt this number of training steps as the default setting for reporting efficiency.
>
> **Exp. 5. Ablation Study on training recipes (number of frames).**
>
> We selected 81 frames as a balanced trade-off.
> - It provides sufficient temporal duration to capture complete distinct biological events (e.g., cell division) while maintaining high temporal coherence and manageable computational costs.
> - Additionally, 81 frames is a native, optimal temporal window for the Wan2.1[1].
>
> > **Question 1: Typos.**
>
> **AQ1**: Thank you for your careful reading. We have corrected these typos and thoroughly check the manuscript.
> - Figure 1: "**Mircoscale**" -> "Microscale".
> - Figure 3 "**Boraders**" -> "Borders".
>
> > Reference
>
> - [1]: Wan T, Wang A, Ai B, et al. Wan: Open and Advanced Large-Scale Video Generative Models. arXiv preprint arXiv:2503.20314, 2025.

---

### Official Review · Reviewer_qfnB · 2025-10-30

**Soundness:** 3
**Presentation:** 3
**Contribution:** 3
**Rating:** 6
**Confidence:** 3

**Summary:**

This paper introduces MicroVerse, a preliminary exploration toward micro-world simulation, aiming to extend video generation models to capture biologically and physically meaningful microscale dynamics (e.g., organ-, cellular-, and subcellular-level processes). The authors propose MicroWorldBench, a rubric-based benchmark with 459 expert-annotated tasks for evaluating microscale simulations, and build MicroSim-10K, a large-scale dataset of 9.6K physics-grounded video clips. Based on this dataset, they fine-tune Wan2.1 to obtain MicroVerse, which reportedly improves scientific fidelity compared to existing open-source video generation models. The paper highlights a new research direction and provides valuable resources for future study.

**Strengths:**

- The paper opens an underexplored and impactful direction — using video generation models for microscale (biomedical) simulation, which connects generative AI with scientific and educational applications.

- MicroWorldBench and MicroSim-10K are well-constructed and carefully validated, providing a foundation for future research in scientific video generation.

-  The rubric-based evaluation (Scientific Fidelity, Visual Quality, Instruction Following) combined with expert refinement offers a transparent and interpretable assessment framework.

-  The paper is well-written and conceptually coherent, linking macroscopic “world models” with microscopic phenomena convincingly.

**Weaknesses:**

- MicroVerse is essentially a fine-tuned version of Wan2.1 without introducing new architectural or physical modeling components. The improvement in scientific fidelity (+0.8 overall) is marginal.

- While the benchmark and dataset are significant contributions, the proposed model does not clearly outperform commercial systems or strong open-source baselines, suggesting that the key contribution lies in data curation rather than modeling innovation.

- The scoring process relies heavily on GPT-5 grading, which may introduce subjectivity or bias. The human validation is limited in scale and lacks quantitative rigor.

- Despite the “physics-grounded” claim, no explicit physical constraints or differentiable physical priors are used in the model training, which limits its scientific credibility.

**Questions:**

- How scalable is the rubric-based evaluation if new tasks or domains are added? Would each require renewed expert verification?
- Is there any plan to integrate explicit physical priors (e.g., fluid or diffusion equations) to improve scientific fidelity?
- How consistent are GPT-5 rubric scores compared to human experts across diverse biological processes?
- Can the authors clarify how much of the performance improvement is due to dataset quality versus model fine-tuning?

---

> ### Author Response · Authors · 2025-11-24
> **Reply to Reviewer qfnB, Part 1**
>
> We sincerely appreciate your valuable feedback. Below, we address each concern.
>
> > **Weakness 1: MicroVerse is essentially a fine-tuned version of Wan2.1 without introducing new architectural or physical modeling components. The improvement in scientific fidelity (+0.8 overall) is marginal.**
>
> **A1**: We acknowledge that it does not introduce new architectural or physical modeling components, but our goal is to provide a **proof of concept** showing that **physics- and biology-aware training data can meaningfully improve scientific fidelity in microscale simulation**. The "marginal" +0.8 overall improvement masks a substantial targeted gain in Scientific Fidelity, which specifically improves from 40.3 to 43.0 on the 1.3B model.
>
> The modest overall average is diluted by slight, expected decreases in Visual Quality and Instruction Following as the model shifts toward a highly specialized domain. To demonstrate that mixing general-domain data effectively preserves the above scores, we randomly sampled an equal amount of data from OpenVid[1], a high-quality dataset in general-domain scenarios, and combined it with MicroSim-10K for fine-tuning. The results are summarized below.
>
> |Variant|Data Size|Scientific Fidelity|Visual Quality|Instruction Following|
> |:-|:-|:-|:-|:-|
> |Baseline Model (Wan2.1-1.3B)|0|40.3|71.8|50.1|
> |FT on MicroSim-10K|9,601|43.0|68.5|49.3|
> |FT on general-domain data and MicroSim-10K|19,202|**44.1 (+3.8)**|74.4 (+2.6)|53.8 (+3.7)|
>
> > **Weakness 2: The model doesn’t clearly beat commercial or strong open-source baselines, implying the main contribution is data curation, not modeling innovation.**
>
> **A2**: The performance gap in our initial submission was primarily due to insufficient model scale (1.3B). During the rebuttal, we scaled our approach to Wan2.1-14B. As shown table below, with adequate capacity and mixed-domain training, **our model now outperforms existing open-source baselines, achieving a +5.6 gain in Scientific Fidelity**.
>
> |Variant| Data Size | Scientific Fidelity| Visual Quality | Instruction Following|
> |:-|:-| :- |:-|:-|
> | Baseline Model (Wan2.1-14B) | 0 | 42.7 | 86.0 | 53.8 |
> | FT on MicroSim-10K | 9,601 | 45.4 | 82.7 | 51.4 |
> | FT on general-domain data (Randomly sample data from OpenVid[1]) and MicroSim-10K | 19,202 | **48.3 (+5.6)** | 87.7 (+1.7) | 56.9 (+3.1) |
>
> Moreover, **the SOTA performance we achieve among open-source models effectively meets the critical need for on-premise deployment**. Since commercial APIs typically restrict usage to cloud environments, MicroVerse serves as an open-source alternative that enables researchers to deploy and customize microscale simulations within their own infrastructure, eliminating the need to rely on external APIs.
>
> > **Weakness 3: The scoring process relies heavily on GPT-5 grading, which may introduce subjectivity or bias. The human validation is limited in scale and lacks quantitative rigor.**
>
> **A3**:  Thank you for raising this point. Our results in *Figure 6(b)* show that GPT-5’s rubric-based scoring broadly follows the same trends as human evaluators.
>
> To further address concerns about potential subjectivity in GPT-5’s judgments, we measured Fleiss’ Kappa for a human-only evaluation panel (3 of our experts) and for a panel that additionally includes GPT-5 (3 humans + GPT-5). The agreement scores are as follows:
>
> | Evaluation Group         | Agreement Score |
> |:--------------------------|:-----------------:|
> | Human-only (Fleiss' Kappa) | 0.733           |
> | Human + GPT-5 (Fleiss' Kappa) | 0.771         |
> | GPT-5 vs Gemini-2.5-pro (Spearman) | 0.835     |
> | GPT-5 vs GPT-5 (Spearman)  | 0.912          |
>
> These results suggest that adding GPT-5 does not introduce additional subjective noise, and its scoring consistency is even better than that of human experts.
>
> > **Weakness 4: Despite the “physics-grounded” claim, no explicit physical constraints or differentiable physical priors are used in the model training, which limits its scientific credibility.**
>
> **A4**:  We acknowledge the "physics-grounded" claim is not well-implemented. We apologize that some statements may have been confusing or slightly overclaimed.
>
> We rephrased them as below:
> - line 23-24: we construct MicroSim-10K, a high-quality, ~~physically-grounded~~ $\color{red}{\text{expert-verified}}$ simulation dataset.
> - line 90-91: MicroSim-10K, the first ~~physically grounded~~ $\color{red}{\text{microscale}}$ dataset containing 9,601 expert-verified scenarios.
> - line 98: we construct MicroSim-10K, a large-scale, expert-verified ~~physically grounded~~ dataset of microscale simulation videos.
> - line 289-290: Toward Microscale Simulation via a ~~Physics-Grounded~~ $\color{red}{\text{Expert-Verified}}$ Dataset.

---

> ### Author Response · Authors · 2025-11-24
> **Reply to Reviewer qfnB, Part 2**
>
> > **Question 1: How scalable is the rubric-based evaluation if new tasks or domains are added? Would each require renewed expert verification?**
>
> **AQ1**:  We acknowledge that rubric-based evaluation is, in principle, not scalable. It does require renewed expert verification.
>
> - ****Our framework could reduces the expert workload.****
>
> However, when new tasks are added, our framework substantially reduces the expert workload. It does this by leveraging GPT-5’s strong prior knowledge to automatically generate fine-grained rubric drafts. These drafts include criterion descriptions, polarity, and importance weights, all created before experts step in. As described in Section 2.3 and Appendix G, experts only perform lightweight revision and validation (e.g., removing redundant criteria, adjusting weights, or adding a small number of missing scientific requirements) rather than creating rubrics from scratch. This design shifts the high-volume, low-ambiguity work to LLMs and retains only the final scientific check for domain experts.
>
> In practice, for a task with \~15 rubric items, an expert only needs to skim the reference video, review the LLM-generated rubric, and make minor edits—taking about **3–5 minutes**, compared to **\~20 minutes** required to manually craft rubrics from scratch. Thus, our pipeline both preserves expert-level reliability and significantly improves scalability for expanding to new tasks.
>
> - ****Potencial to be scalable for rubric-based evaluation.****
>
> To further explore scalability, we tested our rubric-based evaluation on a **RAG QA task** using a **subset of 20 queries** from the TREC DL dataset. Specifically, we adapted rubrics from the *Pencils Down!* [2] framework as **few-shot examples**.
>
> The results showed a **Spearman correlation** of 0.70 between GPT-5-generated scores and human volunteers’ scores, with **89%** of the rubrics deemed appropriate by the volunteers:
>
> | Domain    | #Queries | #Rubric items / query | Corr. (GPT-5 vs human) | Accuracy |
> |:-----------|:----------:|:-----------------------:|:----------------------:|:----------:|
> | RAG-QA    | 20       | 9.2                   | 0.70 Spearman        | 0.89     |
>
> While these results are promising, they are not yet perfect. We believe that with continued advances in LLMs and prompt engineering, the scalability of rubric-based evaluation across different domains will improve, leading to even better results in the future.
>
> > **Question 2: Is there any plan to integrate explicit physical priors (e.g., fluid or diffusion equations) to improve scientific fidelity?**
>
> **AQ2**:  **Yes! Integrating explicit physical priors is a key direction for our future work.**
>
> We fully agree with the reviewer that while our current data-driven approach captures visual dynamics effectively, embedding explicit physical constraints is crucial for achieving high-precision scientific fidelity. We are currently exploring methods to integrate explicit physical priors into our models.
>
> - First, we are exploring how to filter and synthesize a dataset of MicroWorld simulations that adheres to relevant motion equations, such as those governing fluid dynamics and molecular transport. We expect this dataset to enable the model to learn the prior knowledge of microscopic physical motion.
>
> - Second, we plan to integrate physics-informed loss constraints into our diffusion training process[3][4][5], such as the Navier-Stokes equation for fluids or the Advection-Diffusion equation for molecular transport. These constraints are intended to penalize model outputs that violate fundamental physical laws, thereby improving the scientific fidelity of the generated results.
>
> By combining the generative power of MicroVerse with these explicit physical priors, we believe we can significantly enhance the scientific reliability of microscale simulations.
>
> > References
>
> - [1]: Nan K, Xie R, Zhou P, et al. Openvid-1m: A large-scale high-quality dataset for text-to-video generation[J]. arXiv preprint arXiv:2407.02371, 2024.
> - [2]: Farzi N, Dietz L. Pencils down! automatic rubric-based evaluation of retrieve/generate systems[C]//Proceedings of the 2024 acm sigir international conference on theory of information retrieval. 2024: 175-184.
> - [3]: Bastek J H, Sun W C, Kochmann D M. Physics-informed diffusion models[J]. arXiv preprint arXiv:2403.14404, 2024.
> - [4]: Agarwal, N., et al. Cosmos World Foundation Model Platform for Physical AI. arXiv preprint arxiv:2501.03575, 2025.
> - [5]: Chu, M., et al. Learning Temporal Coherence via Self-Supervision for GAN-based Video Generation. arXiv preprint arXiv:1811.09393, 2020.

---

> ### Author Response · Authors · 2025-11-24
> **Reply to Reviewer qfnB, Part 3**
>
> > **Question 3: How consistent are GPT-5 rubric scores compared to human experts across diverse biological processes?**
>
> **AQ3**:  We present the following table showing Fleiss' Kappa for human-only and human + GPT-5 panels across three levels' biological processes:
>
> | Level            | Agreement (Human-only) | Agreement (Human + GPT-5) |
> |:------------------|:------------------------:|:---------------------------:|
> | Organ-level      | 0.728                  | 0.742                     |
> | Cellular-level   | 0.740                  | 0.739                     |
> | Subcellular-level| 0.691                  | 0.727                     |
>
> These results suggest that GPT-5’s consistency across biological processes is comparable, or even superior, to human evaluators. However, at the **Subcellular-level**, both human experts and GPT-5 show some subjectivity, indicating areas where there might still be variation in their judgments.
>
> > **Question 4: Can the authors clarify how much of the performance improvement is due to dataset quality versus model fine-tuning?**
>
> **AQ4**: We conducted an ablation study comparing fine-tuning on our curated MicroSim-10K versus a larger, uncleaned raw dataset. The results are summarized below.
>
> |Variant|Data Size|Scientific Fidelity |Visual Quality |Instruction Following |
> |:-|:-|:-|:-|:-|
> |Baseline Model (Wan2.1-1.3B)|0|40.3|71.8|50.1|
> |FT on raw (uncleaned) data|34,318|42.7|67.4|44.5|
> |FT on MicroSim-10K|9,601|43.0|68.5|49.3|
>
> Key Findings:
> - **Quality Over Quantity**: Fine-tuning on the curated MicroSim-10K achieves higher Scientific Fidelity (43.0 vs. 42.7) using only 28% of the data volume compared to the raw dataset.
> - **Conclusion**: While fine-tuning triggers the domain adaptation, dataset quality is the decisive factor for achieving superior scientific fidelity and training efficiency.

---

> > ### Comment · Reviewer_qfnB · 2025-11-25
> > **Official Comment by Reviewer qfnB**
> >
> > Thank you for your constructive feedback and for evaluating the additional experiments. I maintain my rating.

---

### Official Review · Reviewer_Snf5 · 2025-10-31

**Soundness:** 2
**Presentation:** 2
**Contribution:** 1
**Rating:** 0
**Confidence:** 5

**Summary:**

This paper introduces MicroWorldBench, a rubric-based benchmark designed to evaluate microscale simulation tasks across multiple biological and physical scales, including organ-level dynamics, cellular processes, and molecular interactions. The benchmark includes 459 expert-annotated evaluation criteria covering dimensions such as scientific accuracy, temporal consistency, visual quality, and instruction following. The authors show that current state-of-the-art video generation models perform poorly on these tasks, exhibiting physical law violations and biologically inaccurate behavior. To address this gap, they create MicroSim10K, an expert-verified dataset of physically grounded microscale simulations, and train MicroVerse, a video generation model specialized for this domain. MicroVerse demonstrates improved fidelity in reproducing complex microscale phenomena. Overall, the work introduces the concept of "micro-world simulation" and presents an initial benchmark, dataset, and model aimed at advancing scientific and biological video simulation capabilities.

**Strengths:**

This work creates a curated microscale simulation dataset (MicroSim-10K) with expert verification, addressing lack of domain-specific training data.

Presents a fine-tuned model (MicroVerse) that improves fidelity and consistency on the proposed benchmark, showing feasibility of domain adaptation.

**Weaknesses:**

By looking at Figure 1, my primary concern is that the simulations themselves do not appear realistic (as these simulations look they were made for educational purposes). In some cases, the model-generated outputs look more realistic than the ground-truth simulations (e.g., the SORA cell-division example). This raises questions about whether the benchmark’s “expert-verified” reference simulations accurately reflect real **biological phenomena** and whether the evaluation metric truly measures **biological fidelity** rather than stylistic similarity (because again, these are not realistic life cell imaging videos, rather educational style videos).

I believe this work could still be framed as generating educational biological simulations or illustrative biology videos, but claiming true biological fidelity is not supported by the examples shown (anyone with basic wetlab experience can discern that these videos are not realistic). The current outputs do not convincingly reflect real biological processes, making it difficult to justify the biological realism claims.

Given these facts  (and that I don't think it is possible to make a single claim of biological fidelity) I recommend either expanding this dataset to include real biological videos or restructuring the whole paper.

**Questions:**

Please address this main concern above ^

---

> ### Author Response · Authors · 2025-11-22
> **Reply to Reviewer Snf5, Part 1**
>
> We sincerely appreciate your valuable feedback. Below, we address each concern.
>
> > **Weaknesse 1: By looking at Figure 1, my primary concern is that the simulations themselves do not appear realistic (as these simulations look they were made for educational purposes). In some cases, the model-generated outputs look more realistic than the ground-truth simulations (e.g., the SORA cell-division example).This raises questions about whether the benchmark’s “expert-verified” reference simulations accurately reflect real biological phenomena and whether the evaluation metric truly measures biological fidelity rather than stylistic similarity (because again, these are not realistic life cell imaging videos, rather educational style videos).**
>
> **A1**: We acknowledge your observation that *“the model-generated outputs look more realistic than the ground-truth simulations”*. However, we clarify that our claim of **biological fidelity refers to mechanistic correctness rather than visual realism**. The reference simulations collected from YouTube are mostly educational in style, and they have been reviewed by our experts to help confirm that the underlying biological mechanisms are accurate. Therefore, we believe that *"the benchmark’s “expert-verified” reference simulations do accurately reflect real biological phenomena"*.
>
> To further support this point, we conducted an additional independent evaluation of the reference simulations. We randomly sampled 20 educational style clips that were **not** part of the original benchmark tasks. For each clip, we created a new task–rubric pair using our rubric framework and one of our experts reviewed them. Two of our experts then scored the clips, and we took the average. **As shown in below Table, the reference simulations reach a scientific fidelity score of 87.4, higher than all evaluated models**. In contrast, the Sora-generated videos appear visually realistic but may not fully reflect real biological phenomena.
>
> > Table. Sora's high visual realism does not correlate with scientific fidelity.
>
> |Model|Scientific Fidelity |Visual Quality |
> |:-|:-:|:-:|
> | HunyuanVideo | 15.6 | 48.2 |
> | CogVideoX-5B | 37.4 | 64.1 |
> | Wan2.1-T2V-1.3B | 40.3 | 71.8 |
> | Wan2.2-TI2V-5B | 40.7 | 82.7 |
> | Wan2.1-T2V-14B | 42.7 | 86.0 |
> | Wan2.2-T2V-A14B | 37.8 | 92.8 |
> | MicroVerse-1.3B (Ours) | 43.0 | 68.5 |
> | Veo3 | 65.7 | 97.0 |
>  **Sora** | **35.3** | **96.4** |
> | **Reference simulations (20 sampled clips)** | **87.4** | **96.1** |
>
> We argue that **MicroWorldBench does measure scientific fidelity rather than stylistic similarity**. We conducted two additional style-control experiments to verify that the benchmark is not biased toward stylistic similarity.
>
> **Exp. 1 Scientific Fidelity Remains Stable Across Styles in Reference Simulations.**
>
> We also sampled 20 realistic-style clips from the reference simulations to form a balanced comparison set. Using these 20 realistic and 20 educational clips, we find that Scientific Fidelity remains very close across styles. The results are summarized below.
>
> |Style Type|Scientific Fidelity |Visual Quality |
> |:-|:-:|:-:|
> |Realistic Style|**89.9**|97.0|
> |Educational Style|**87.4**|96.1|
>
> **Exp. 2 Style Does Not Affect Scientific Fidelity in Model-Generated Outputs.**
>
> Using 200 prompts, we generated both realistic style and educational style outputs using Sora and report the averaged scores. Scientific Fidelity remains stable despite large appearance changes. The results are summarized below.
>
> |Sora Output Style|Scientific Fidelity |Visual Quality |
> |:-|:-:|:-:|
> |Realistic Style|**36.1**|96.8|
> |Educational Style|**35.4**|88.1|
>
> These results show that **style does not meaningfully change Scientific Fidelity, even though it causes large variations in visual appearance, confirming that the benchmark measures biological fidelity rather than stylistic similarity**. We will include these analyses in the revision and scale up the study.

---

> ### Author Response · Authors · 2025-11-22
> **Reply to Reviewer Snf5, Part 2**
>
> > **Weaknesse 2: I believe this work could still be framed as generating educational biological simulations or illustrative biology videos, but claiming true biological fidelity is not supported by the examples shown (anyone with basic wetlab experience can discern that these videos are not realistic). The current outputs do not convincingly reflect real biological processes, making it difficult to justify the biological realism claims**.
>
> **A2**: We acknowledge that the work can be framed as generating educational biological simulations. The reference simulations have been reviewed by our experts to help confirm that they capture biological phenomena rather than merely exhibiting stylistic similarity.
>
> We also apologize that some statements may have been confusing or slightly overclaimed.
> We rephrased them as below:
>
> - line 23-24: we construct MicroSim-10K, a high-quality, ~~physically-grounded~~ $\color{red}{\text{expert-verified}}$ simulation dataset.
> - line 90-91: MicroSim-10K, the first ~~physically grounded~~ $\color{red}{\text{microscale}}$ dataset containing 9,601 expert-verified scenarios.
> - line 98: we construct MicroSim-10K, a large-scale, expert-verified ~~physically grounded~~ dataset of microscale simulation videos.
> - line 289-290: Toward Microscale Simulation via a ~~Physics-Grounded~~ $\color{red}{\text{Expert-Verified}}$ Dataset.
>
> We also added several new clarifications, shown below:
> - line 29: $\color{blue}{\text{Our work demonstrates the potential of educational microscale simulations of biological mechanisms.}}$
> - line 184: $\color{blue}{\text{Scientific fidelity emphasizes mechanistic accuracy rather than visual realism.}}$
> - line 488-490: $\color{blue}{\text{Our work aims to explore the potential of educational microscale simulations of biological mechanisms, rather than the reproduction of results }}$
> $\color{blue}{\text{observed in wet lab experiments.}}$
>
> Therefore, we revise it using blue color, see the updated version.
>
> > **Weaknesse 3: Given these facts (and that I don't think it is possible to make a single claim of biological fidelity) I recommend either expanding this dataset to include real biological videos or restructuring the whole paper.**
>
> **A3**: Regarding your constructive suggestion to either expand the dataset or restructure the paper, we have taken both steps.
>
> - **First, we’ve added real biological videos in the revisedd revision,** although the available data remain limited. Using the method described in *Section 3.1*, we collected 377 real biological videos from YouTube and **obtained 643 video clips** after preprocessing. *Figure 9 in Appendix E* shows an example of the newly added read biological video clips. We then combined these data with MicroSim-10K for fine-tuning. The results are summarized below.
>
> |Variant|Data Size|Scientific Fidelity|Visual Quality|Instruction Following|
> |:-|:-|:-:|:-:|:-:|
> |Baseline Model (Wan2.1-1.3B)|0|**40.3**|71.8|50.1|
> |FT on MicroSim-10K|9,601|**43.0**|68.5|49.3|
> |FT on MicroSim-10K and real biological video clips|10,244|**41.7**|67.6|49.8|
>
> - **Second, we have restructured the paper.** Specifically, we (i) clarified the framing to emphasize that the goal is educational biological simulation rather than realistic microscale simulation; (ii) clarified the wording around biological fidelity to avoid giving the impression of a single claim and to make clear that we refer to mechanistic correctness within educational simulations; and (iii) reorganized the introduction and discussion to better align contributions, limitations, and intended use cases with the dataset’s characteristics.
>
> Thank you for the helpful suggestions, which have helped us improve the clarity and positioning of the work.

---

> > ### Comment · Reviewer_Snf5 · 2025-11-26
> >
> > I apologize if I missed this, but throughout the manuscript and the response, experts are frequently mentioned. Could you please provide information on each expert’s field of expertise and years of experience?
> >
> > **Response to A1:** The table does not address my main concern. Since the simulations do not appear realistic, I would expect the use of established distributional metrics (e.g., Fréchet Inception Distance, FID) to quantitatively compare real biological videos with the simulations. Without such analyses, it is difficult to assess whether the simulated data matches the real data distribution.
> >
> > I am also concerned that the proposed evaluation criteria are biased toward idealized simulations rather than real biological phenomena. For example, the “Visual Quality” criterion emphasizes  "high clarity and fine presentation of microstructural details, with sharp edges of subcellular structures, well-defined layer separation, and absence of wax artifact noise". Whereas real microscopy images (e.g. electron microscopy)  are inherently noisy and variable (e.g., due to shot noise, specimen properties, and detector limitations), and organelles often appear less sharply defined. Thus, if a model produces accurate, realistic simulations, the proposed evaluation schema would heavily penalize it. This is exacerbated by the fact that the evaluation is done by LLMs. Indeed, even SOTA LLMs are terrible at understanding  real microscopy [1].
> >
> > [1] Burgess, James, et al. "Microvqa: A multimodal reasoning benchmark for microscopy-based scientific research." Proceedings of the Computer Vision and Pattern Recognition Conference. 2025.
> >
> > The results of Experiment 1 are also surprising. Given the criteria listed in the appendix, it is difficult to believe that real and simulated videos could achieve similar scores. For example, the criterion “textures are credible” may hold for real videos, but not for simulations.  Could you please also report FID for this experiment?  I  also recommend adding human experts to run this evaluation and calculate the correlation of experts vs LLMs.
> >
> >
> >
> > **Response to A2 and A3:**
> > Thank you for alleviating my concerns. I believe this work remains a valuable contribution; however, the claims should be more carefully scoped. Given the remaining issues with A1, it is difficult to support the claim that the method “captures biological phenomena.” A more accurate description may be that it captures biological mechanisms as represented by simulations.
> >
> > I acknowledge that my concern is specific, but it is important to avoid overclaiming. My main concern is that, if accepted with the current wording, researchers may rely on this benchmark to advance video-based microscopy models under the assumption that it reflects real microscopy distributions. In reality, the simulations remain far from true experimental distributions, and I strongly doubt that models trained solely on these simulations would generalize to real biological data. A useful analogy is that even first-year biology students often struggle to identify organelles in real microscopy images, despite clear textbook schematics, because real experimental data look substantially different. I expect AI models to face similar challenges when trained only on idealized simulations.
> >
> > In light that some of my concerns have been addressed, I will raise my score. I am willing to raise my score more if  A1 is addressed correctly.  Thanks

---

> > > ### Author Response · Authors · 2025-11-28
> > > **Second Reply to Reviewer Snf5, Part 1**
> > >
> > > We once again sincerely appreciate your valuable feedback. Below, we address each concern.
> > >
> > > > **Weakness 1: Could you please provide information on each expert’s field of expertise and years of experience?**
> > >
> > > **A1**: **All experts hold doctoral degrees (Ph.D.) and have extensive research experience in cellular and molecular biology**. Below, we provide the background information for each expert involved in our evaluation:
> > >
> > > |Expert ID|Field of Expertise|Years of Experience|
> > > |:-:|:-|:-:|
> > > |1|Cell Biology/Genetics|12 yeas|
> > > |2|Biochemistry / Metabolic Pathway Modeling|10 years|
> > > |3|Biochemistry / Metabolic Pathway Modeling|8 years|
> > >
> > > > **Weakness 2: The table does not address my main concern. Since the simulations do not appear realistic, I would expect the use of established distributional metrics (e.g., Fréchet Inception Distance, FID) to quantitatively compare real biological videos with the simulations. Without such analyses, it is difficult to assess whether the simulated data matches the real data distribution.**
> > >
> > > **A2**: Following your valuable recommendation, we compute **Fréchet Video Distance (FVD)** as a video-extension of FID. Prior biomedical video synthesis work reports FVD in the 10^2 range (≈80–300) for high-fidelity diffusion/GAN models [1,2].
> > >
> > > > [1] Nguyen V P, et al. *Training-free condition video diffusion models for single frame spatial-semantic echocardiogram synthesis*. MICCAI 2024.
> > >
> > > > [2] Algethami N, et al. *Generative AI for biomedical video synthesis: a review*. Artificial Intelligence Review, 2025.
> > >
> > > #### FVD comparison across models (lower = closer distribution)
> > >
> > > | Data | FVD vs. MicroSim-10K ↓ | FVD vs. 643-Real-Biological-Clips ↓ |
> > > |---|---:|---:|
> > > | MicroSim-10K | 0 | 123.9 |
> > > | 643-Real-Biological-Clips | 123.9 | 0 |
> > > | **Commercial Models** |||
> > > | Veo3 | **42.6** | **118.1** |
> > > | Sora | 116.9 | 136.3 |
> > > | **Open-Source Models** |||
> > > | Wan2.1-T2V-1.3B | 83.0 | 158.9 |
> > > | Wan2.2-TI2V-5B | 77.6 | 153.3 |
> > > | Wan2.1-T2V-14B | 53.3 | 137.6 |
> > > | Wan2.2-T2V-A14B | 65.8 | 132.2 |
> > > | MicroVerse-1.3B (Ours) | 63.6 | 142.7 |
> > > | **MicroVerse-14B (Ours)** | **50.28** | **128** |
> > >
> > > The **FVD** between MicroSim-10K and real biological videos is only **123.9**, indicating that our *expert-verified MicroSim-10K already lie remarkably close to the real microscopy distribution in terms of visual statistics and structural descriptors*, even though they may appear “less realistic” to the naked eye. **We hope this quantitative result directly addresses your concern of unrealism.**
> > >
> > > Furthermore, MicroVerse reduces FVD on both the educational simulations and the real microscopy clips, outperforming all open-source baselines. This consistent improvement **supports our claim that expert-verified data enables the model to better capture biological mechanisms**, thereby narrowing the gap to real experimental videos.

---

> > > ### Author Response · Authors · 2025-11-28
> > > **Second Reply to Reviewer Snf5, Part 2**
> > >
> > > > **Weakness 3: I am also concerned that the proposed evaluation criteria are biased toward idealized simulations rather than real biological phenomena. For example, the “Visual Quality” criterion emphasizes "high clarity and fine presentation of microstructural details, with sharp edges of subcellular structures, well-defined layer separation, and absence of wax artifact noise". Whereas real microscopy images (e.g. electron microscopy) are inherently noisy and variable (e.g., due to shot noise, specimen properties, and detector limitations), and organelles often appear less sharply defined. Thus, if a model produces accurate, realistic simulations, the proposed evaluation schema would heavily penalize it. This is exacerbated by the fact that the evaluation is done by LLMs. Indeed, even SOTA LLMs are terrible at understanding real microscopy.**
> > >
> > > **A3**: We acknowledge that our current evaluation criteria lean toward idealized simulations. Our goal is **educational, mechanism-focused visualization**, where clear structure helps convey biological processes. The emphasis on sharp boundaries, clean layer separation, and low artifact noise **reflects failure modes we frequently observe in existing video generators**, such as structure collapse, blurred organelles, and layer merging. These rubrics were distilled from expert-curated failure cases.
> > >
> > > We also understand your concern about realism being penalized. As noted in our previous rebuttal, we had already performed an independent human-evaluation using **20 realistic-style clips**. Reviewing the results again, we found that **real videos were not downgraded for biological noise or variability** and most deductions came instead from subjectivity in human judgement.
> > >
> > > | Style Type               | Scientific Fidelity | Visual Quality | Evaluator |
> > > | ------------------------ | :------------------: | :-------------: | --------- |
> > > | 20-Realistic-Style-Clips |            **89.9** |       **97.0** | Human     |
> > >
> > > However, as video generation models continue to improve, your concern may indeed become valid. We therefore plan to further expand and refine our rubric to better reward accurate yet noisy realism in future work.
> > >
> > > > **Weakness 4: The results of Experiment 1 are also surprising. Given the criteria listed in the appendix, it is difficult to believe that real and simulated videos could achieve similar scores. For example, the criterion “textures are credible” may hold for real videos, but not for simulations. Could you please also report FID for this experiment?**
> > >
> > > **A4**: We would like to clarify the design of Experiment 1. Two experts generated tasks and rubrics based on the **reference simulation clips**, and a different expert then scored those **same clips** using the rubrics. Therefore the high and similar scores are expected. *The purpose of this experiment was to show that Scientific Fidelity evaluates mechanistic accuracy rather than stylistic realism.*
> > >
> > > Following your suggestion, we report **FVD** between the **20 realistic-style clips** used in Exp.1 and our newly expanded set of **643 expert-verified real biological videos**, against the **Cellular-level subset of MicroSim-10K**. We report FVD at the cellular level because **both real datasets contain ≥90% cellular clips**, while organ- and subcellular-level microscopy data are far less available.
> > >
> > > Directly comparing the 20-realistic and 20-educational clips from Exp.1 would **not** be meaningful, as they were scored under **different** tasks and rubric pairs.
> > >
> > > | Comparison Pair| FVD vs. MicroSim-10K (Cellular)↓  |
> > > | - | :-: |
> > > | 20-Realistic-Style-Clips                  |                    **116.9** |
> > > | 643-Real-Biological-Clips  |                    **132.2** |
> > >
> > > These values fall in the 10² scale, indicating that MicroSim-10K is quantitatively close to real microscopy distributions at the cellular level.

---

> > > ### Author Response · Authors · 2025-11-28
> > > **Second Reply to Reviewer Snf5, Part 3**
> > >
> > > > **Weakness 5: I also recommend adding human experts to run this evaluation and calculate the correlation of experts vs LLMs.**
> > >
> > > **A5**: As we addressed previously in our rebuttal to reviewer **qfnB**, we have already conducted human expert scoring and expert–LLM correlation analysis. Three domain experts independently evaluated the videos, and we quantified scoring agreement between (1) a human-only panel and (2) a panel including GPT-5. Results are shown below:
> > >
> > > | Evaluation Group                   | Agreement Score |
> > > | ---------------------------------- | :--------------: |
> > > | Human-only (Fleiss' Kappa)         |       **0.733** |
> > > | Human + GPT-5 (Fleiss' Kappa)      |       **0.771** |
> > > | GPT-5 vs Gemini-2.5-pro (Spearman) |       **0.835** |
> > > | GPT-5 vs GPT-5 (Spearman)          |       **0.912** |
> > >
> > > We further report agreement across biological scales:
> > >
> > > | Level             | Agreement (Human-only) | Agreement (Human + GPT-5) |
> > > | ----------------- | :---------------------: | :------------------------: |
> > > | Organ-level       |              **0.728** |                 **0.742** |
> > > | Cellular-level    |              **0.740** |                 **0.739** |
> > > | Subcellular-level |              **0.691** |                 **0.727** |
> > >
> > > Overall, GPT-5 shows agreement with expert scoring at a level comparable to human evaluators, and in some cases its consistency is even slightly higher. This suggests that GPT-5 is a reliable evaluator rather than a source of additional variance.
> > >
> > > > **Weakness 6: however, the claims should be more carefully scoped. Given the remaining issues with A1, it is difficult to support the claim that the method “captures biological phenomena.” A more accurate description may be that it captures biological mechanisms as represented by simulations.**
> > >
> > > **A6**: Thank you for this precise suggestion. Following your guidance, we carefully revised all relevant statements in the paper to describe *biological mechanisms* rather than *biological phenomena*, as shown below:
> > > - line 026: MicroVerse can accurately reproduce complex microscale ~~phenomena~~ $\color{red}{\text{mechanism}}$.
> > > - line 092~093: Unlike human-scale datasets, MicroSim-10K emphasizes physical plausibility and biological fidelity across diverse microscale ~~phenomena~~ $\color{red}{\text{mechanisms}}$.
> > > - line 170~171: This information was then provided to GPT-4o, which generated tasks describing the microscale ~~phenomena~~ $\color{red}{\text{mechanism}}$.
> > > - line 293: The results of MicroWorldBench indicate that current models remain limited in their ability to model microscale ~~phenomena~~ $\color{red}{\text{mechanism}}$ governed by physical and biological principles.
> > > - line 351: The dataset spans diverse biological ~~phenomena~~ $\color{red}{\text{mechanisms}}$ across organ, cellular, and subcellular levels, offering broad coverage of key scenarios.
> > >
> > > All revisions see the updated version.
> > >
> > > **We sincerely appreciate your detailed and insightful feedback throughout this review process. Your comments greatly improved the precision and rigor of our work. We hope that the additional analyses and revisions have fully addressed your concerns, and we thank you again for helping us strengthen this paper!**

---

### Author Response · Authors · 2025-12-03
**Summary of Rebuttal**

Dear Area Chair,

Thank you for the extra time and effort during this challenging period. Below we summarize the main revisions and the progress made during the rebuttal process.

> **Key Revisions and Clarifications Based on Reviewer Feedback**

- **Clarification of Scope and Claims.** We clarified that the goal of the work is *educational simulations of biological mechanisms*, not realistic microscopy reproduction. All statements that could be interpreted as overclaims were carefully rewritten to emphasize mechanistic correctness rather than realism.
- **New Quantitative Analyses.** We added multiple new evaluations, including (i) a style-controlled analysis demonstrating that the Scientific Fidelity dimension of our benchmark reflects mechanistic correctness rather than visual style; and (ii) **a new Fréchet Video Distance (FVD) analysis comparing MicroSim-10K, real microscopy videos, and our MicroVerse outputs, directly addressing the core concerns about realism and distributional alignment.**
- **Expert Information and Validation.** We provided detailed background (fields, years of experience) for all experts involved in rubric verification, and we expanded our human–LLM agreement analysis to show that GPT-5 aligns closely with human evaluators.
- **Model Scaling and Ablation Study.** We introduced a 14B MicroVerse model that achieves competitive performance with strong open-source baselines. We further added ablations on filtering strategies, dataset size, CFG rates, and training steps, showing that curated data quality drives scientific fidelity.

> **Reviewer Responses and Rating Changes**

We engaged in extensive and constructive discussion with reviewer *Snf5* (initial score: 0). After our major clarifications and new analyses, the reviewer *Snf5* **raised the rating** and explicitly **expressed a willingness to raise it further**:

*“In light that some of my concerns have been addressed, I will raise my score. I am willing to raise my score more if A1 is addressed correctly.”*

In our second replies, we reported an FVD analysis showing that MicroSim-10K lies quantitatively close to real microscopy videos in distribution, **directly addressing the remaining concerns in A1 about realism and distributional alignment.** We also added further human–LLM agreement analyses showing that our rubric-based evaluation is robust and consistent with expert judgments.

For the other two reviewers (*qfnB, 9ESj*), we carefully addressed all of their concerns with new experiments and clearer explanations. Both reviewers maintained positive evaluations, keeping their scores at 6.

**Overall**, our initial ratings were 0/6/6. The reviewer *Snf5* who originally gave the score of 0 has already raised this score (but invisible) and has indicated an intention to raise it further.

We sincerely thank all reviewers for their valuable comments, and we are deeply grateful for your time and consideration in assessing our submission.

Sincerely,
The Authors

---

### Meta-Review · Area_Chair_7Zh9 · 2026-01-07

**Summary:**

The reviewers' main concerns are the following:

1. Reviewer Snf5 raised a critical concern that the simulations themselves do not appear realistic (as these simulations look they were made for educational purposes). The reviewer recommended either expanding this dataset to include real biological videos or restructuring the whole paper.

2. The scoring process relies heavily on GPT-5 grading, which may introduce subjectivity or bias. The human validation is limited in scale and lacks quantitative rigor (reviewer qfnB). Similar concerns are raised by reviewer 9ESj.

3. Despite the “physics-grounded” claim, no explicit physical constraints or differentiable physical priors are used in the model training, which limits its scientific credibility (reviewer qfnB).

4. Both the training set (MicroSim-10K) and test set (MicroWorldBench) are built entirely from YouTube videos, which may not reflect the full range of scientific simulation requirements. The authors have not reported how they deduplicate to ensure the test set is fully separate from the training set. Additionally, there is a concern that private models (like Veo3) may have already been trained on all YouTube videos. (reviewer 9ESj).

5. Missing ablations. No ablation studies on how dataset filtering stages, dataset size, or training recipes (CFG rate, steps, frames) affect the final performance.

6. Furthermore, I don't find any content in the anonymous repository https://anonymous.4open.science/r/rsrsyzyz/README.md at the time of meta-review.

**Reviewer Concerns:**

After the rebuttal, most of the critical concerns are properly addressed. The authors are required to incorporate the full suggestions into the camera-ready, as well as open-source the dataset and the evaluation. Overall the paper has contributed valuable benchmark into the community.

**Reviewer Scores:**

After the rebuttal, reviewer Snf5 may increase the score from 0 to 4. The other two reviewers have stated that they remain their score of 6.

---

### Decision · Program_Chairs · 2026-01-26

Accept (Poster)